# Individual differences in belief updating and phasic arousal are related to psychosis proneness
Peter R. Murphy [1,2] ✉, Katarina Krkovic[3], Gina Monov[1], Natalia Kudlek[1], Tania Lincoln [3,5] & Tobias H. Donner [1,4,5] ✉

Many decisions entail the updating of beliefs about the state of the environment by accumulating noisy sensory evidence. This form of probabilistic reasoning may go awry in psychosis. Computational theory shows that optimal belief updating in environments subject to hidden changes in their state requires a dynamic modulation of the evidence accumulation process. Recent empirical findings implicate transient responses of pupil-linked central arousal systems to individual evidence samples in this modulation. Here, we analyzed behavior and pupil responses during evidence accumulation in a changing environment in a community sample of human participants. We also assessed their subclinical psychotic experiences (psychosis proneness). Participants most prone to psychosis showed overall less flexible belief updating profiles, with diminished behavioral impact of evidence samples occurring late during decision formation. These same individuals also exhibited overall smaller pupil responses and less reliable pupil encoding of computational variables governing the dynamic belief updating. Our findings provide insights into the cognitive and physiological bases of psychosis proneness and open paths to unraveling the pathophysiology of psychotic disorders.

Many decisions entail the sequential accumulation of noisy, incomplete, or ambiguous pieces of information about the state of the world – a form of probabilistic reasoning that we here refer to as dynamic belief updating. This process is particularly challenging in natural environments[1], which mix multiple sources of uncertainty such as noise degrading the evidence obtained during a stable environmental state, and the possibility of hidden changes in the environmental state itself[2]. Aberrant belief updating yields biased and/or noisy representations of the environment, false expectations about future events, and consequently maladaptive patterns of cognition and behavior evident in many mental disorders including psychosis[3,4].

Indeed, problems in constructing accurate beliefs are a hallmark of psychotic experiences, such as delusions (i.e., fixed beliefs contradicting the evidence) or hallucinations (non-veridical percepts). While these experiences are key symptoms of patients diagnosed with schizophrenia, they are also reported by people in the general population[5]. An emerging line of research points to aberrations of probabilistic reasoning in psychosis[3,6–9], giving rise to specific cognitive biases. For example, one well-documented bias in psychosis is the tendency to commit early to a particular interpretation of incoming information (also termed evidence) without assessing all that is available, often referred to as 'jumping to conclusions' bias[10,11]. Another, and so far disjunct, line of research has established a dysregulation (typically, increase) of tonic levels of physiological arousal in psychosis[12–16]. Whether and how changes in arousal levels in individuals with psychotic experiences relate to aberrations of probabilistic reasoning—in particular, to dynamic belief updating – remain open questions.

Here, we developed an integrated approach to close this gap, building on recent computational and cognitive neuroscience studies implicating central arousal systems in dynamic belief updating[17–19]. A growing body of work on various behavioral tasks with noisy evidence and change-points in the environmental state has established that human participants often approximate the normative (i.e., accuracy-/reward-maximizing) evidence accumulation strategy[17,19–23]. This normative process entails a dynamic 'upweighting' of new evidence (or, analogously, 'down-weighting' of prior beliefs) at moments when either the probability of a change in the hidden state, or uncertainty about the hidden state, are high[2,17,19,24]. Such transient increases in evidence sensitivity at the expense of prior beliefs are, in part, mediated by transient ('phasic') responses of ascending neuromodulatory systems, such as the locus coeruleus norepinephrine system[25,26]. In most

[1]Section Computational Cognitive Neuroscience, Department of Neurophysiology and Pathophysiology, University Medical Center Hamburg-Eppendorf, Hamburg, Germany. [2]Department of Psychology, Maynooth University, Co. Kildare, Ireland. [3]Department of Clinical Psychology and Psychotherapy, Institute of Psychology, University of Hamburg, Hamburg, Germany. [4]Bernstein Center for Computational Neuroscience, Charité Universitätsmedizin, Berlin, Germany. [5]These authors contributed equally: Tania Lincoln, Tobias H. Donner. ✉e-mail: peter.murphy@mu.ie; t.donner@uke.de

studies of phasic arousal responses during of dynamic belief updating, across both visual and auditory domains, the activity of these systems has been inferred from the dilation of the pupil[17,19,24], an established marker of central arousal state[27–32].

We reasoned that this recruitment of phasic arousal during dynamic belief updating may be disturbed in psychosis, and that this disturbance might link the tonic hyper-arousal found in psychotic individuals[12–16] to their characteristic reasoning biases[3,6–9]. Specifically, because of a well-documented opponent interplay between tonic and phasic activity modes of central arousal systems[25], we assumed that increased tonic arousal levels should reduce phasic, task-evoked arousal responses during cognition[33]. If phasic arousal responses fail to track the probability of hidden state changes ('change-point probability', CPP) or the agent's uncertainty during belief updating as a result, this should reduce the dynamic evidence re-weighting in changing environments, altering the dynamic belief updating process overall and potentially leading to more perseverative, inflexible beliefs. To test these hypotheses, we here related individual differences in the proneness to psychosis to the dynamics of evidence accumulation in a volatile environment, as well as to the associated pupil-linked arousal responses.

## Methods

### Recruitment and sample

The study was approved by the ethics committee of the Faculty of Psychology and Movement Science at Universität Hamburg and included a total of 96 human participants. To avoid overly strong skewness of P-scores with the majority scoring at the low end of the psychosis continuum, we aimed to recruit a sample that mirrored distributions found in large-scale community samples, where the 50th percentile equalled a P-score of 1.4[34]. An interim check of our sample data after recruiting 73 participants revealed the low end of the continuum to be slightly over-represented. We therefore adjusted our recruiting strategy for remaining participants by pre-screening for P-scores above 1.4. Out of 75 pre-screened individuals, 36 reached this cutoff, of which 18 completed the study. Additionally, 5 randomly chosen individuals with P-scores lower than 1.4 were invited to participate. As a result, we arrived at a sample with a distribution that corresponds to the distributions found in more representative community samples[34]. All participants provided written informed consent, were aged between 18–55 years old, fluent in the German language, had an IQ > 85 as assessed by the Multiple Choice Vocabulary Test[35], had normal/corrected-to-normal vision, were not pregnant, and had no dementia or other organic brain disorders, acute intoxication or acute suicidal tendencies. Participants received remuneration in the form of an hourly rate (€10 per hour), a bonus for completing both of two planned sessions (€15), and compensation if they incurred costs for a SARS-CoV-2-antigen test to participate in the study.

Four participants were excluded from all analyses due to failing to complete both testing sessions, and 2 more participants were excluded due to incomplete CAPE questionnaire responses. The remaining 90 participants (mean ± s.d. age of 31.6 ± 9.5 years, range 18–55; 48 female participants and 42 male participants; both age and sex determined via self-report; data on race/ethnicity not collected) were included in all analyses of data from the main behavioral task, having completed two experimental sessions (session 1: 180 min; session 2: 160 min). A further 2 participants were excluded from analyses of data from the delayed match-to-sample task having failed to complete any blocks of this task due to time constraints.

Of the 90 participants included in all main analyses, four self-reported a previous diagnosis of depression, one of emotionally unstable personality disorder (impulsive type), one of post-traumatic stress disorder (PTSD), one of obsessive-compulsive disorder (OCD), one of comorbid depression and attention-deficit hyperactivity disorder (ADHD), and one of psychosis due to substance abuse. The latter participant had a CAPE P-score of 2.1, which placed them in the highest P-score quintile. Except for the summed modulations of evidence weighting by CPP and -|ψ| (see 'Psychophysical kernels' below; effect trending at $P = 0.1$ with this participant excluded), all effects of P-score quintile reported in the main text remained statistically significant even when this participant was excluded from the analysis.

For participants that completed the full experimental protocol, the first session consisted of study information and consent, brief intelligence screening via Multiple Choice Vocabulary Test[35] and visual attention and task switching task Trail-Making-Test (TMT)[36], a questionnaire battery, training on the main behavioral task and 7–8 experimental blocks of this task (see below). The questionnaire battery was administered after the first three experimental blocks, after which the remaining experimental blocks took place. The second session consisted of 8-9 blocks of the main behavioral task as well as training and between 2 and 3 experimental blocks of the delayed match-to-sample working memory task halfway through the set of blocks of the main task (see below). We also measured at-rest heart rate variability at the beginning of each session via electrocardiogram, though results pertaining to these data are not reported here.

### Questionnaire battery

The Community Assessment of Psychic Experiences (CAPE)[37,38] is a self-report questionnaire consisting of 42 items that assesses frequency and distress of lifetime psychotic experiences. It subdivides into three factors: negative symptoms, positive symptoms, and depression symptoms. Previous research has shown that the CAPE has strong evidence of both convergent and discriminative validity, as demonstrated by Hanssen et al.[39]. Additionally, the instrument has been shown to be reliable over time, with good test-retest reliability[37]. The German version of the CAPE, which we employed here, has also been validated in terms of its factorial and criterion validity[34]. Here, we focused on the frequency dimension of the P-score, both in regard to sampling the high end of the continuum and in terms of the main analyses. The D- and N- scores were used to assess specificity of the findings.

The Trier Inventory for Chronic Stress (TICS)[40] and Trauma History Questionnaire (THQ)[41] were used to assess the chronic stress load of the participants. Self-efficacy was assessed with the General Self-Efficacy Scale (SWE)[42] and achievement motivation with the Achievement Motives Scale-Revised (AMS-R)[43]. Due to the fact that these scales were utilized to address hypotheses unrelated to the present study, we do not describe them in detail here nor report related results.

### Main behavioral task

The main task was a two-alternative forced choice task in which the generative task state $S = \{left, right\}$ could change unpredictably (Fig. 1a). Participants were asked to maintain fixation at a centrally presented mark throughout the trial, monitor a sequence of evidence samples and report their inference about $S$ at the end of the sequence.

Stimuli were generated using Psychtoolbox 3 for Matlab[44]. Visual stimuli were presented in a behavioral laboratory on a Dell P2210 22-inch monitor with resolution set to 1680 x 1050 and refresh rate to 60 Hz. Participants were seated with their head in a chinrest 52 cm from the monitor during task performance.

Stimuli were presented against a gray background. Three placeholders were present throughout each trial: a light-gray vertical line extending downward from fixation to 7.4° eccentricity; a colored half-ring in the lower visual hemifield (polar angle: from -90 to +90° relative to bottom of vertical meridian; eccentricity: 8.8°) which depicted the $LLR$ associated with each possible sample location; and a fixation mark as a black disc of 0.18° diameter superimposed onto a disk of 0.36° diameter with varying color informing participants about trial intervals. The colors comprising this half-ring and the fixation point were selected from the Teufel colors[45]. Evidence samples consisted of achromatic, flickering checkerboards (temporal frequency: 10 Hz; spatial frequency: 2°) within a circular aperture (diameter = 0.8°), and varied in polar angle (constant eccentricity of 8.1°).

Samples were presented for 250 ms (sample-onset asynchrony (SOA): 400 ms). Samples were centered on polar angles $x_1,…,x_n$ drawn from one of two truncated Gaussian distributions $p(x \mid S)$ with variance $\sigma_{left} = \sigma_{right} = 27°$ and means symmetric about the vertical meridian ($\mu_{left} = -17°$, $\mu_{right} = +17°$). These parameters equated to a signal-to-noise ratio (SNR) for the generative distributions of $(\mu_{right}-\mu_{left})/\sigma = 1.26$. If a draw $x_i$ was <-90°

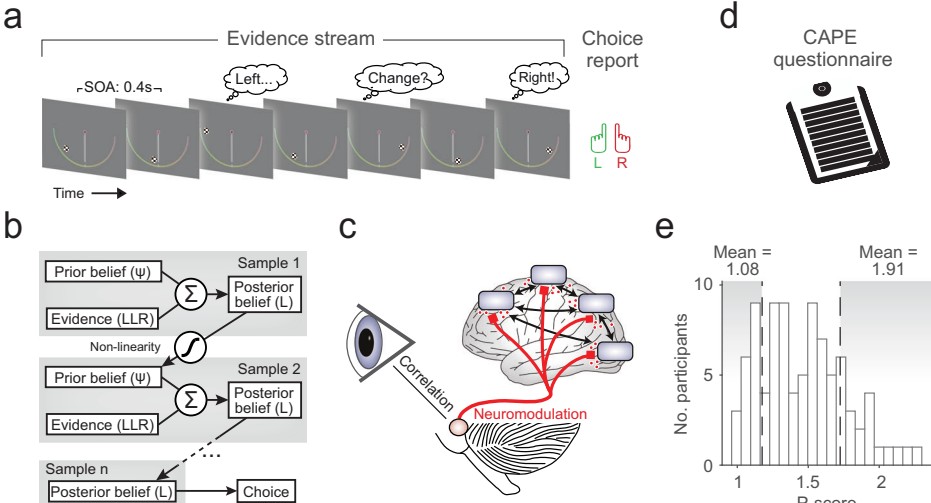

**Fig. 1 | Behavioral task and approach. a** Schematic of two-alternative perceptual choice task with hidden state changes. Locations of successive evidence samples (checkerboard patches) were drawn from one of two noisy sources that switched unpredictably over time, and participants reported the inferred active source when the sequence terminated. Each possible sample location provided a unique amount of evidence for one or the other alternative, as illustrated by the color bar shown to participants throughout (yellow ≈ no evidence, green ≈ strong evidence for leftward source, red ≈ strong evidence for rightward source). The sample stimulus-onset asynchrony (SOA) was 400 ms. **b** Schematic of normative belief updating process.

**c** Tracking central arousal state through monitoring of pupil diameter. Rapid, non-luminance-mediated dilations of the pupil are an established proxy of the activity of neuromodulatory brainstem systems with wide projections throughout the brain, by which they control cortical network state. See main text for details. **d** Measuring individual psychosis proneness via CAPE questionnaire. **e** Distribution of psychosis proneness (P-scores) extracted from questionnaire data (*n* = 90 participants). Vertical dashed lines indicate cutoffs for lowest and highest P-score quintiles, determining subgroups (*n* = 18 participants in each) used for the majority of our analyses; means are mean P-scores within each subgroup. **a, b** Adapted from ref. 17.

(>+90°), it was replaced with −90° (+90°). *S* was chosen at random at the start of each trial and could change with a hazard rate $H = p(S_n = right \mid S_{n-1} = left) = p(S_n = left \mid S_{n-1} = right) = 0.1$ after each sample. 65% of sequences contained 10 samples. The remaining 35% (randomly distributed in each block) contained (with equal probability) 2-9 samples and were introduced to encourage participants to attend to all samples. These task parameters were chosen to ensure expressive belief updating dynamics characterized by periods of strong belief coupled with regular moments of surprise and uncertainty. The motivation behind specific parameter values is provided in Supplementary Table 1.

Each trial began with a variable fixation baseline period (uniform between 0.5–1.5 s) during which a stationary checkerboard patch was presented at 0° polar angle. This checkerboard then began to flicker, followed after 400 ms by the first evidence sample. The final sample of the sequence was then replaced by the stationary 0° patch and a 'Go' cue instructed participants to report their choice 0.7–1.2 s (uniform) after sequence end. They pressed left or right 'CTRL' keys on a keyboard, with their left or right index fingers, to indicate *left* or *right*, respectively. Auditory feedback 0.05 s post-response informed participants whether their choice corresponded to the true *S* at sequence end (see *Stimuli*): an ascending 350 → 950 Hz tone (0.25 s) for correct choice and descending 950 → 350 Hz tone for incorrect choice. This was followed by an inter-trial interval of 1.2 s when participants could blink. During the preparatory interval, sample sequence and subsequent delay, the color of the second fixation disk was light red; the 'Go' cue was this disk becoming light green; and, the inter-trial period was indicated by the disk becoming light blue.

Participants practiced the task prior to the main experiment, with detailed initial illustrations of the generative distributions and incremental increases in trial complexity. These instructions were delivered using a cover story that the task involved viewing sequences of footballs (the evidence samples) that two players kicked from the same point (fixation) but aiming at different goals (polar angles corresponding to the means of the two generative distributions). The players had quite bad aim (variance of the generative distributions) and could swap in and out of play between each kick with a fixed probability (the task *H*); and the participants' task was to infer which player kicked the last football in each sequence. They were never informed about the exact task *H*, rather that the likelihood of a change in football player (*S*) from one sample to the next was "not very high. For example, there will be some trials in which there are no changes at all. However, many trials will have one change, and some trials can even have several changes." Following training, participants completed 15-16 blocks of 86 trials of the task, spread over both testing sessions. They received feedback about their average performance at the end of each block, and were instructed to fixate the central disk and minimize blinking during the trial.

## Working memory task

Participants also completed a visuospatial delayed match-to-sample working memory task[46] presented on the same monitor as the inference task, again via Psychtoolbox 3. The task was to decide whether a sample stimulus and a test stimulus separated by a variable delay occurred in the same or different locations. Each trial began with presentation of a central white fixation cross (arm length: 0.8 degrees of visual angle, d.v.a.; arm thickness: 0.2 d.v.a.) that was present for the entire trial. After a variable baseline interval (uniform distribution with range 0.5-2.0 s), the sample stimulus was presented for 0.5 s, followed by the delay (1, 3 or 9 s, equiprobable) and then the test stimulus (0.5 s). Sample and test stimuli were circular checkerboard patches (diameter: 2.8 d.v.a.; spatial frequency: 1 cycle per d.v.a.), appearing in the lower visual hemifield at a fixed eccentricity of 6 d.v.a. The sample could be presented at any of 12 equiprobable locations, ranging from ~-76.15° to ~76.15° of polar angle (fixed spacing ≈ 13.85°), while the most extreme samples could still be flanked by a 'near non-match' test stimulus on both sides (amounting to 14 possible test stimulus locations, spaced evenly from −90° to 90°). The test occurred at either the same location as the sample or at a different location (see below). Upon offset of the test stimulus, the fixation cross changed color from white to light blue, which prompted participants to report their decision via right- or left-handed button press for "same" or "different" judgments, respectively. This response was soon (0.1 s) followed by visual feedback about its accuracy ("Correct" in green font; "Error" in red font; font size 36, presented 1.0 d.v.a. above fixation for 0.75 s). Each trial was followed by a fixed 2 s interval during which participants were instructed to blink if needed, and this was followed by the baseline period of the following trial.

The task was designed to consist of three trial categories, each with a desired frequency of occurrence within a block of trials: 'Match trials' (sample and test at identical positions, 1/3 of trials), 'Near non-match trials' (smallest possible sample-test distance of 13.85°; 1/3 of trials), and 'Far non-match trials' (sample-test distance randomly chosen from the remaining possible sample-test distances, which could be between 27.7° and ~166.15° depending on the sample location; 1/3 of trials). Trials were presented in blocks of 63 trials each, within which the different delay durations and sample-test distances were randomly interleaved under the above-mentioned constraint. Participants received feedback about their average performance at the end of each block. They were instructed to fixate the central cross and minimize blinking during the trial.

Participants also underwent initial training to familiarize them with the task. This consisted of a general instruction of the task rules through PowerPoint slides, as well as practice with various aspects of the task (stimulus appearance, timings, response contingencies) that, as above, grew in complexity culminating in practice of full task trials. Once training was complete, each participant then performed 2 or 3 blocks of the task (mean = 2.25, s.d. = 0.44), with the specific number dependent on time constraints.

Trials on which participants pressed a key other than the designated response keys, response time was ≤ 0.2 s, or response time exceeded the participant's mean response time by 4 s.d. were all excluded from analysis of response accuracy on the working memory task. Mean response accuracy was computed as the proportion of correct responses (i.e., "same" response on match trials and "different" response on non-match trials).

### Normative model for main behavioral task

The normative model for the main task prescribes the following computation[21]:

$$L_n = \psi_n + LLR_n \tag{1}$$

$$\psi_n = L_{n-1} + \log\left[\frac{1-H}{H} + \exp(-L_{n-1})\right] - \log\left[\frac{1-H}{H} + \exp(L_{n-1})\right] \tag{2}$$

Here, $L_n$ was the observer belief after encountering the evidence sample $x_n$, expressed in log-posterior odds of the alternative task states; $LLR_n$ was the log-likelihood ratio reflecting the relative evidence for each alternative carried by $x_n$ ($LLR_n = \log(p(x_n|right)/p(x_n|left))$); and $\psi_n$ was the prior expectation of the observer before encountering $x_n$. We used this model to derive two computational quantities: $CPP$ and $-|\psi|$. $CPP$ was the posterior probability that a change in generative task state has just occurred, given the expected $H$, the evidence carried by $x_n$, and the observer's belief before encountering that sample $L_{n-1}$, computed as follows (see ref. 17 for derivation):

$$CPP_n = \frac{H(\mathrm{N}(x_n|S_1)p(S_{2,n-1}) + \mathrm{N}(x_n|S_2)p(S_{1,n-1}))}{H(\mathrm{N}(x_n|S_1)p(S_{2,n-1}) + \mathrm{N}(x_n|S_2)p(S_{1,n-1})) + (1-H)(\mathrm{N}(x_n|S_1)p(S_{1,n-1}) + \mathrm{N}(x_n|S_2)p(S_{2,n-1}))} \tag{3}$$

where $\mathrm{N}(x \mid S)$ denoted the probability of sample $x$ given a normal distribution with mean $\mu_S$ and s.d. $\sigma_S$. Uncertainty was defined as the negative absolute of the prior ($-|\psi_n|$), reflecting uncertainty about the generative state before observing $x_n$.

### Main task model fitting and comparison

We first computed the accuracy of participants' choices with respect to the true final generative state, and compared this to the accuracy yielded by three idealized decision processes presented with identical stimulus sequences: normative accumulation (Eqs. 1 and 2), perfect accumulation, and only using the final evidence sample. For each trial, choice $r$ ($left = -1$, $right = +1$) was determined by the sign of the log-posterior odds after observing all samples: $r_{trl} = sign(L_{n,trl})$ for normative, $r_{trl} = sign(\sum_{i=1}^{n} LLR_{i,trl})$ for

perfect, and $r_{trl} = sign(LLR_{n,trl})$ for last-sample, where $n$ indicated the number of samples presented on trial $trl$.

We fit variants of the normative model to participants' behavior, assuming that choices were based on the log-posterior odds $L_{n,trl}$ for the observed stimulus sequence on each trial. In line with previous work with this model[17,21], different model variants had different combinations of the following free parameters: $L_{n,trl}$ was corrupted by a decision noise term $\nu$, such that choice probability $\hat{r}$ was computed as:

$$\hat{r}_{trl} = \frac{1}{2} + \frac{1}{2}\,\mathrm{erf}\left(\frac{L_{n,trl}}{\sqrt{2}\nu}\right) \tag{4}$$

We also allowed for (i) misestimation of the task $H$ by fitting a subjective hazard rate parameter $\hat{H}$; (ii) a bias in the mapping of stimulus location to $LLR$ such that subjective evidence strength $LLR_n = LLR_n \cdot B$, where $B$ captured the equivalent of under- (B < 1) or over-estimation (B > 1) of the signal-to-noise ratio of the task generative distributions (also known as 'expected uncertainty'[2]); and (iii) a bias in the weighting of evidence samples that were (in)consistent with the existing belief, such that $LLR_n = LLR_n \cdot IU$ for any sample $n$ where $sign(LLR_n) \neq sign(\psi_n)$.

We fitted the parameters by minimizing the cross-entropy between participant and model choices:

$$e = -\sum_{trl}(1 - r_{trl})\log(1 - \hat{r}_{trl}) + r_{trl}\,\log(\hat{r}_{trl}) \tag{5}$$

where $r_{trl}$ was the participant choice. This objective was minimized via particle swarm optimization (PSO toolbox[47]), setting wide bounds on all parameters and running 300 pseudorandomly-initialized particles for 1500 search iterations. The relative goodness of fit of different model variants was assessed by the Bayes Information Criterion (BIC):

$$BIC = 2e + k\log(n) \tag{6}$$

where $k$ was the number of free parameters and $n$ was the number of trials.

We fit four different model variants including various combinations of the above free parameters (free parameters in each variant are listed in Supplementary Fig. 2a). Model validation through parameter recovery analyses served to rule out two models that included the $IU$ parameter capturing a bias in weighting of (in)consistent samples, because these models had substantially inflated variance of $\hat{H}$ estimates (Supplementary Fig. 2b, c) that were strongly co-linear with the $IU$ estimates themselves (Supplementary Fig. 2d), revealing significant identifiability issues. Quantitative model comparison (based on BIC scores derived from Eq. 6; Supplementary Fig. 2e) of the two remaining model variants favored a simple 2-parameter normative model fit that included only decision noise ($\nu$) and subjective hazard rate ($\hat{H}$) as free parameters.

### Psychophysical kernels

We quantified the time course of the impact of evidence from the main task on choice using logistic regression:

$$\mathrm{logit}[p(r_{trl} = right)] = \beta_0 + \sum_{i=1}^{n}\beta_{1,i} \cdot LLR_{i,trl} + \sum_{j=2}^{n}(\beta_{2,j} \cdot LLR_{j,trl} \cdot CPP_{j,trl} + \beta_{3,j} \cdot LLR_{j,trl} \cdot (-|\psi|_{j,trl})) \tag{7}$$

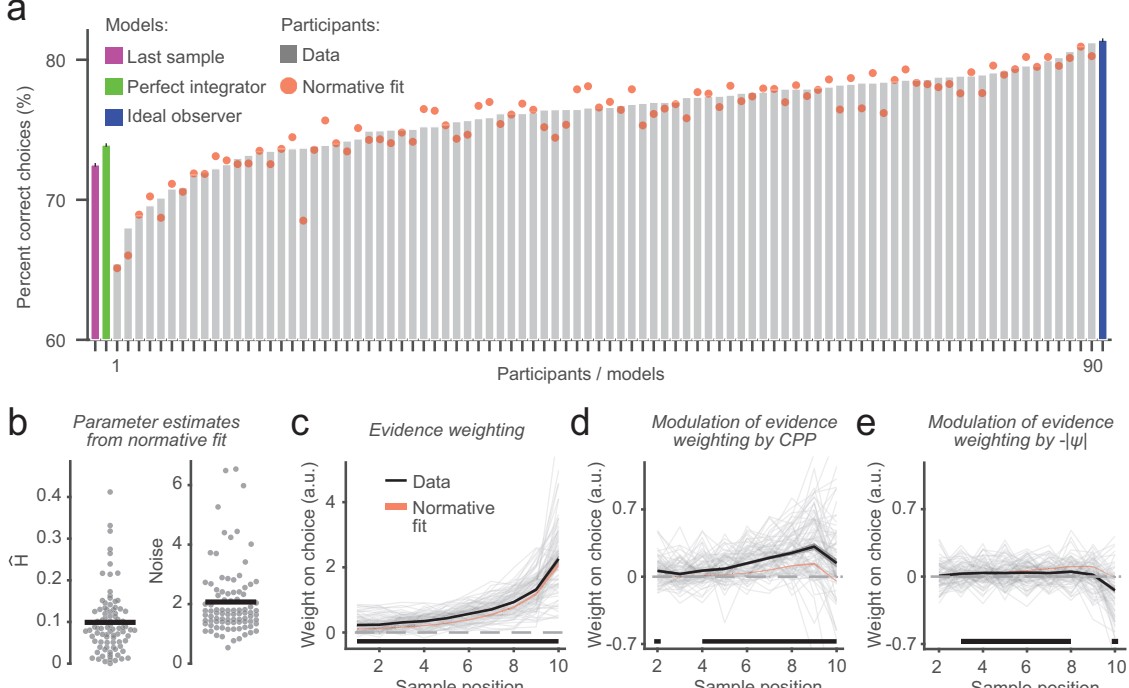

**Fig. 2 | Behavioral signatures of dynamic belief updating. a** Choice accuracies of $n = 90$ human participants (gray bars), with accuracies of the ideal observer (navy), perfect integrator (green) and deciding based on last sample heuristic (green) given the same stimulus sequences shown for reference. Red dots, accuracies of normative model fits. **b** Best-fitting parameters of normative model fits for individual participants (gray circles) and group means (black horizontal bars). **c** Time-resolved weight of evidence on choice, derived by regressing participants' single-trial choices onto evidence strengths (log-likelihood ratios, LLRs) per sample position within each trial. **d, e** Modulation of evidence weighting, estimated via LLR interaction terms in the regression model, by two variables that the normative model prescribes sensitivity to: change-point probability (CPP, **d**), which captures the degree to which a new evidence sample conflicts with the current belief; and uncertainty ($-|\psi|$, **e**) in the belief prior to observing each new sample. In **c–e**: black, mean and s.e.m. (shaded areas) of participants' data; significance bars, time points where weights differ from zero ($P < 0.05$, two-tailed cluster-based permutation test; evidence weighting and largest CPP and $-|\psi|$ modulation clusters, $P < 0.0001$). Red shadings, mean $+/-$ s.e.m. of normative model fits. Gray thin lines, individual participants. $n = 90$ participants for calculation of means and s.e.m. in all panels.

where $i$ and $j$ indexed sample position within sequences of $n$ samples ($n = 10$ for this analysis), and $LLR$ was the true $LLR$. The dependent variable was participants' choice (left = 0, right = 1) or model choice probability. The set $\beta_1$ quantified the impact of evidence at each position on choice and the sets $\beta_2$ and $\beta_3$ modulations of evidence weighting by $CPP$ and $-|\psi|$, respectively. $CPP$ was logit-transformed, and both variables were then z-scored. Additionally, all regressors were z-scored across the trial dimension. We fit the logistic regression models using 10-fold cross-validation and an L1 regularization term of $\lambda = 0.002$, without which convergence issues were encountered for a small number of participants.

We tested for sample positions with above-chance evidence weighting ($\beta_1$) and $CPP$ ($\beta_2$) and $-|\psi|$ ($\beta_3$) modulations by comparing the group-level distributions of beta weights against zero using cluster-based permutation testing (two-tailed; 10,000 permutations; cluster-forming threshold: $P < 0.01$), which corrects for multiple comparisons across sample positions[48]. For each of the $CPP$ and $-|\psi|$ modulations, this procedure yielded a large contiguous cluster of sample positions characterized by the expected positive effects (between sample positions 4–10 for $CPP$, and sample positions 3–8 for $-|\psi|$; Fig. 2c–e). For analyses of individual differences (across-participant correlations, comparison of sub-groups defined by CAPE scores), we averaged the beta weights within these clusters to compute a scalar modulation score per individual participant; and for such analyses combining $CPP$ and $-|\psi|$ modulations, we summed across these two scalar values. We computed a 'kernel half difference' measure capturing the relative degree of primacy or recency in average evidence weighting by subtracting the mean of $\beta_{1,1-5}$ from the mean of $\beta_{1,6-10}$. We also divided this measure by the sum across all $\beta_1$ to compute a complementary normalized kernel half difference measure that is independent of individual differences in overall kernel magnitude[49].

Differences in kernel-based measures across participant subgroups (e.g., lowest and highest quintiles) defined by CAPE P-scores were assessed via two-sample permutation test (two-tailed; 10,000 permutations). Confidence intervals around computed statistics were estimated via bootstrapping with replacement (10,000 permutations), and we quantified evidence for or against the hypothesis of an effect of subgroup through Bayes factors ($BF_{10}$; using the Cauchy distribution with scale parameter $\sqrt{2}/2$ as the prior on standardized effect sizes[50]). Across-participant correlations were assessed via Spearman correlation.

**Pupillometry acquisition and preprocessing**

Gaze position and pupil diameter from both eyes were recorded at 250 Hz with an SMI RED 500 Eye-Tracker. We restricted our current analysis to pupil data from the main behavioral task. For each task block, we first selected the eye (left *vs* right) giving rise to the pupil signal with the lowest variance across the entire time series and discarded the signal from the other eye from further analysis. Blinks and noise transients were removed from the remaining pupillometric time series using a linear interpolation algorithm in which artifactual epochs were identified via thresholding of the raw pupil size (1.5 mm) and the first derivative of the z-scored time series (threshold = $\pm 3.5$ zs$^{-1}$). The average time series was then band-pass filtered (0.06–6 Hz, Butterworth), re-sampled to 50 Hz, and z-scored per block. We computed the first derivative of the result, referred to as *pupil* below. Finally, any trial (from onset of the first evidence sample to 5.5 s after trial onset) in which either left or right pupil signals were contaminated by >60% artifactual samples, or any sample forming part of a contiguous artifactual epoch of longer than 1 s, was excluded from all further analysis. For visualization and analysis of the trial-related pupil response in the raw signal (but not it's derivative), we baseline-corrected the signal for each

trial by subtracting the mean pupil size in a 0.1 s window centered on that trial's onset.

## Modeling of evoked pupil responses

We assessed the sensitivity of *pupil* to computational variables by segmenting the signal from 0–1 s after sample onset (full-length trials) and fitting:

$$pupil_{t,s,trl} = \beta_0 + \sum_{g=s-1}^{s}(\beta_{1,g} \cdot CPP_{g,trl} + \beta_{2,g} \cdot (-|\psi|_{g,trl}) + \beta_{3,g} \cdot |LLR|_{g,trl})$$
$$+ \beta_4 \cdot x\_gaze_{t,s,trl} + \beta_5 \cdot y\_gaze_{t,s,trl} + \beta_6 \cdot base_{s,trl}$$

$$(8)$$

where $t$ indicated time point, $x\_gaze$ and $y\_gaze$ were horizontal and vertical gaze positions, and $base$ was the 'baseline' pupil diameter (−0.05–0.05 s around sample onset). $|LLR|$ captured a possible relationship with 'unconditional' Shannon surprise[17]. Previous sample $CPP$, $-|\psi|$ and $|LLR|$ were included because the pupil response is slow, meaning correlations with variables from the previous sample may have caused spurious effects.

We tested for sample-aligned time points with above-chance $CPP$ ($\beta_1$) and $-|\psi|$ ($\beta_2$) encoding via cluster-based permutation test (two-tailed; 10,000 permutations; cluster-forming threshold: $P < 0.01$). For each of the $CPP$ and $-|\psi|$ modulations, this yielded a large contiguous cluster of time points characterized by the expected positive effects (between ~0.4–1.0 s for $CPP$, and ~0.2–0.45 s for $-|\psi|$; Fig. 2c–e). For analyses of individual differences, we averaged the beta weights within these clusters to compute a scalar value capturing encoding strength per individual participant; and for such analyses combining $CPP$ and $-|\psi|$ modulations, we summed across these two scalar values. Informed by visualization of the trajectories of the average trial-evoked pupil response, we also computed two scalar measures capturing the magnitude of this response: by averaging the raw (baselined) signal from 0.5–5.5 s following trial onset, and the derivative from 0.2–1.0 s following trial onset. As above for the analysis of kernels, differences in pupil measures across participant subgroups (e.g., lowest and highest quintiles) defined by CAPE P-scores were assessed via two-sample permutation test (two-tailed; 10,000 permutations), and across-participant correlations were assessed via Spearman correlation.

## Reporting summary

Further information on research design is available in the Nature Portfolio Reporting Summary linked to this article.

## Results

We related individual differences in dynamic belief updating and pupil-linked arousal responses to the individual psychosis proneness in a large community sample of 90 participants (Fig. 1; see Methods for details on recruitment), combining the behavioral task (Fig. 1a), modeling (Fig. 1b) and psychophysiological (Fig. 1c) approach developed in Murphy et al[17]. with responses to an established self-report questionnaire.

Psychosis proneness was assessed with the frequency dimension of the positive (P) score of the Community Assessment of Psychic Experiences (CAPE; Fig. 1d)[37,38]. This score was chosen because it assesses the type of psychotic experiences with an intuitive relationship to aberrant belief formation (i.e., hallucinatory experiences and delusional beliefs), is most indicative of psychosis proneness across the continuum and can be used as a screening tool to identify those at risk of future development of psychosis[51,52]. In line with the recommended scoring procedure, the positive score (henceforth: P-score) was calculated by averaging responses across a subset of 20 predefined questionnaire items to compute a single summary metric. We found a broad range of P-scores across our sample, reaching values found in individuals fulfilling established criteria for ultra-high risk of developing psychosis (P-score, frequency dimension > 1.7[52]) in a modest (20%) fraction of participants (Fig. 1e; see Supplementary Fig. 1 for distributions of the N (negative) and D (depressive) scales). With both categorical (e.g., healthy versus diagnosis of schizophrenia or 'ultra-high risk'

status) and continuous (i.e., recognition of a psychosis continuum[5]) views on psychotic symptomatology being prevalent, below we report both discrete (participant binning) and continuous approaches (between-participant correlations) to studying these individual differences. We did not have a priori hypotheses about which, if either, would yield clearer effects. Because the binning approach enabled a direct inspection of the below-described measures of belief updating for different patient sub-groups, we opted for presenting these results in the main figures.

The task (Fig. 1a) required accumulation of evidence provided by discrete visual 'samples', under the possibility of hidden changes in the underlying (categorical) environmental state that participants had to infer. On each trial, between 2 and 10 evidence samples were presented sequentially, with the spatial location of each sample providing noisy information about the hidden state. Sample locations were generated from one of two probability distributions (one per environmental state). The generative state was chosen randomly at the beginning of each trial and could then change after each sample, with a fixed probability (task hazard rate, $H$) of 0.1 in the main experiment (see Supplementary Table 1 for summary of all task parameters and motivation of specific values). Participants were asked to report the generative state at the end of each sequence (left- or right-handed button press).

The normative computation maximizing accuracy on this task (Fig. 1b) entails the accumulation of evidence samples (expressed as log-likelihood ratios, $LLR$, quantifying strength of support for one over the other state) into an evolving belief $L$ (also expressed as log-odds ratio) that governs the final decision[21]. The key difference between the normative model and standard evidence accumulation models (such as drift diffusion[53]) lies in a non-linear function through which the updated belief after each sample $n$ ($L_n$) is passed to compute the prior for the next updating step ($\psi_{n+1}$). This non-linearity depends on the agent's estimate of the environmental volatility, the subjective hazard rate $\hat{H}$. This way, the normative model strikes an optimal balance between formation of strong beliefs in stable environments versus fast change detection in volatile environments. In the following, we used this normative model in its pure form (called 'ideal observer' below, i.e., with exact knowledge of the task $H$ and without any internal noise) as a benchmark against which to compare the behavior of human participants. We also fitted variants of this model combined with deviations from this pure form to each participant's choices.

## Participants accumulate evidence dynamically

We previously found using a combination of approaches—comparison of human decisions to those of idealized decision processes, analysis of specific behavioral readouts of evidence integration, and fitting variants of the normative model to observed choices – that participants base their decisions on our task on the approximately normative belief updating process, subject to internal noise and biased encoding of the environmental statistics. Here, we expected to replicate these findings, applying the same complementary analyses to data from our larger community sample (Fig. 2).

We gained an initial impression of how our participants made decisions on the task by comparing their responses to those of three idealized decision processes, one of which was the ideal observer. While, as expected, all participants achieved accuracies below the theoretical upper limit provided by the ideal observer, most performed better than alternative benchmarks corresponding to sub-optimal decision processes (76.7% with better performance than 'perfect integration', which equates to uniform evidence weighting without information loss and is normative only for completely stable environments, $H = 0$; 87.8% with better performance than a 'choose based on last sample' heuristic which is normative only for completely unpredictable environments, $H = 0.5$) even though these were not contaminated by any noise (Fig. 2a). The participants' choices were also more consistent with those of the ideal observer (same choice on 85.7% of trials, 95% bootstrapped confidence interval (C.I.) = [84.8, 86.6]) compared to the alternative processes (perfect integrator: 78.1%, 95% C.I. = [77.4, 78.9], 95% C.I. on difference of means = [6.4, 8.7], $P < 0.0001$; last-sample heuristic: 76.4%, 95% C.I. = [75.2, 77.7], 95% C.I. on difference of means = [7.8, 10.7], $P < 0.0001$; paired-samples permutation tests).

**Fig. 3 | Change in belief updating profiles for individuals with high psychosis proneness. a** As Fig. 2c, but now for subgroups of participants occupying the lowest (gray) and highest (black) quintiles of P-scores. **b** Kernel half-difference summary measure (subtraction of mean weighting of first 5 samples from mean weighting of last 5 samples) capturing degree of recency in evidence weighting, plotted for both lowest and highest P-score quintiles. **c** As Fig. 2d, e, but now for subgroups occupying lowest and highest P-score quintiles. **d** Summary measure capturing summed strength of modulations of evidence weighting by CPP (mean modulation weights over sample positions 4–10, significant cluster in Fig. 2d) and -|ψ| (mean modulation weights over sample positions 3–8, significant cluster in Fig. 2e), plotted for lowest and highest P-score quintiles. Red shadings, mean +/− s.e.m. of normative model fits. Horizontal greyscale lines in **b** and **d**, mean of data from each subgroup; circles, individual participants. *P*-values, two-sample permutation tests (two-tailed). CI_L-H, 95% confidence intervals (bootstrapped) around difference of means between lowest and highest P-score quintile subgroups. *n* = 18 participants per subgroup for calculation of means, s.e.m. and statistical tests in all panels.

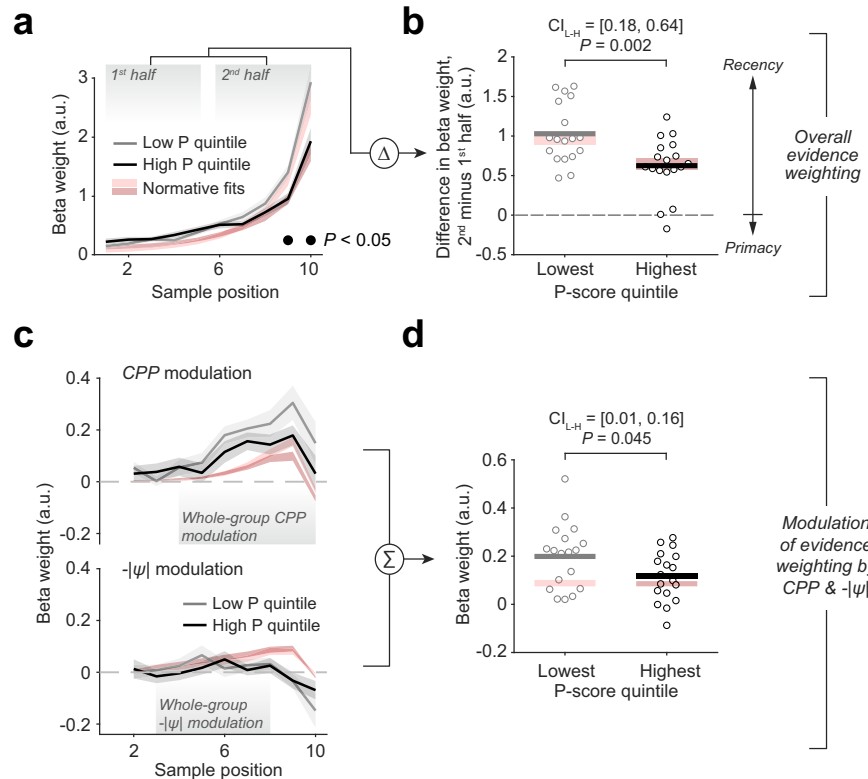

The notion that participants approximated normative belief updating for our volatile environment was also supported by fitting alternative variants of the normative model to their choices (Fig. 2; see *Methods* and Supplementary Fig. 2a for description of candidate models and their free parameters). Informed by model validation and comparison (Supplementary Fig. 2; *Methods*), the variant of the normative model that we used for all further analyses contained two free parameters: a subjective hazard rate ($\hat{H}$) that could deviate from the task hazard rate, and a decision noise parameter[17,21,23] (Fig. 2b). The choices of this best-fitting model variant ('normative fit' in Fig. 2a, red) were highly consistent with participants' choices (86.9%, 95% C.I. = [86.1, 87.7]). The model fits also suggested that while at the group level participants appeared to form accurate estimates of the level of volatility in the task ($\hat{H} = 0.099$, 95% C.I. = [0.084, 0.115]; $P = 0.9$ compared to task H of 0.1, permutation test), there was substantial variation in these estimates across individual participants ($\hat{H}$ range = [0.0003, 0.412]; Fig. 2b). Overall, these results are in line with previous work in smaller samples[17,21,23,54].

We used psychophysical kernels as a model-free psychophysical tool to uncover the temporal profiles of participants' evidence accumulation processes[55] (i.e., evidence weighting profiles). Psychophysical kernels quantify the impact of fluctuations of evidence at a given sequence position on the final choice (Methods). We found significant and increasing weights on choice for all sample positions (all weights above 0), demonstrating that participants used all evidence in their decisions, albeit with a recency temporal profile (i.e., stronger impact for late than early samples; Fig. 2c). This profile is predicted by the normative belief updating model (Fig. 2c, red). We used a simple scalar metric to quantify the recency effect, by subtracting mean weights across the first half of the sequence (first 5 samples) from mean weights across the second half of the sequence (last 5 samples). Positive values of this so-called 'kernel half difference' metric indicated recency in evidence weighting, negative values indicated primacy, and values close to zero indicated equal weight to all samples. The kernel half differences provided clear support for recency (mean = 0.85, 95% C.I. = [0.76, 0.93], $P < 0.0001$, two-tailed permutation test against 0).

Critically, our participants also exhibited modulations of their evidence weighting profiles (Fig. 2d–e) that were previously identified as diagnostic of the normative accumulation process and observed in smaller samples – specifically the dynamic modulation of the impact of evidence on choice by change-point probability (CPP) and uncertainty (-|ψ|)[17]. CPP is the posterior probability of a change in generative state, given the subjective hazard rate, prior belief state and new evidence sample. Because it is contingent on internal belief states and representations of environmental volatility, this latent variable captures a high-level form of surprise. Uncertainty is captured by the (sign-flipped) absolute strength of the belief before a new sample is encountered (i.e., the prior). We estimated CPP and -|ψ|, for each sample *n*, from belief trajectories $L_n$ predicted by the normative fits[17] (Methods) and quantified their modulatory effects on evidence weighting profiles through an extension of our logistic regression approach (Methods).

The CPP modulation can be interpreted as the readiness to change one's belief state in the face of contradictory (disconfirmatory) evidence, a construct prevalent in psychosis research (see Discussion); while the -|ψ| modulation captures a tendency to give greater weight to new information when in a state of uncertainty, often an adaptive policy in changing environments[56,57]. Both modulations reflect the dynamic, context-dependent weighing of new information that permits the flexible belief updating prescribed by the normative model[17]. There was a robust modulation of evidence weighting by CPP ($P < 0.0001$ for cluster encompassing sample positions 4–10, cluster-based permutation test; Fig. 2d), particularly strongly for samples in the second half of the evidence stream (mean CPP 'kernel half difference' = 0.17, 95% C.I. = [0.12, 0.21], $P < 0.0001$, two-tailed permutation test). There was also a reliable -|ψ| modulation ($P < 0.0001$ for cluster encompassing sample positions 3–8; Fig. 2e). As in previous investigations in similar task contexts[17], we found that the effect of CPP was the stronger of these dynamic modulations of evidence weighting (difference of means across each identified cluster = 0.13, 95% C.I. = [0.1, 0.16], $P < 0.0001$, two-tailed permutation test). As also reported previously[17], the observed modulation of evidence weighting by CPP was stronger in the participants than the normative model fit to their choices (Fig. 2d, compare black and red).

**Fig. 4 | Evoked pupil responses and individual differences. a** Average trial-related pupil response (solid line) and its first derivative (dashed line). Dashed gray vertical lines, sample onsets. **b, c** Both measures of overall trial-evoked response for the low and high P-score subgroups. **d** Encoding of change-point probability and uncertainty in pupil responses evoked by individual evidence samples. **e** Encoding of change-point probability and uncertainty in pupil responses (pooled) for low and high P-score subgroups. Shaded areas in (**a, d**) indicate s.e.m. ($n = 90$ participants). Significance bars in **d**, $p < 0.05$ (two-tailed cluster-based permutation test). Horizontal lines in **b, c, e**, mean of data from each subgroup ($n = 18$ participants per subgroup); circles, individual participants. *P*-values, two-sample permutation tests (two-tailed). $CI_{L\text{-}H}$, 95% confidence intervals (bootstrapped) around difference of means between lowest and highest P-score subgroups.

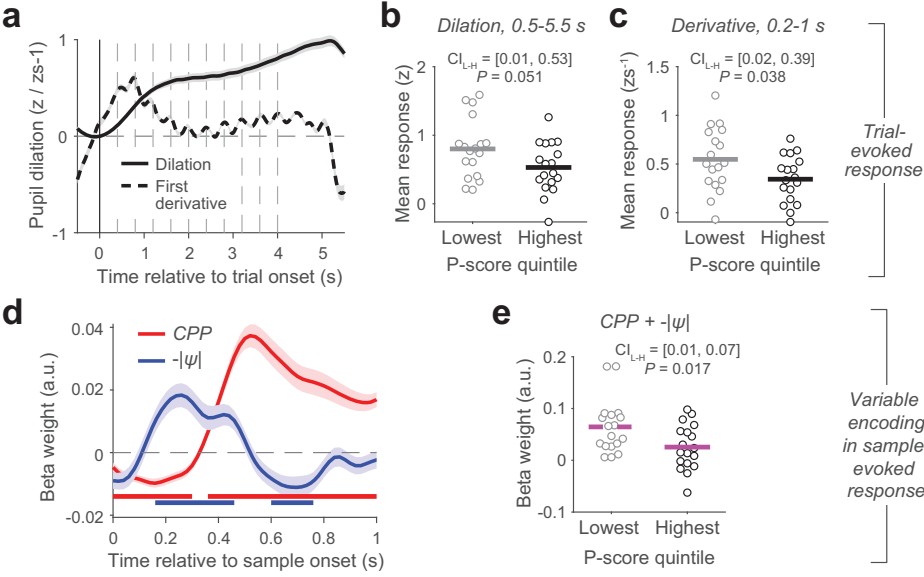

## Participants prone to psychosis exhibit altered signatures of belief updating

One commonly reported cognitive bias in psychosis is the tendency to 'jump to conclusions' – a premature commitment to a particular hypothesis without assessing all the available information[10,11]. A related phenomenon is the so-called 'bias against disconfirmatory evidence' – a failure to change an initial hypothesis about a hidden state in the face of evidence against that hypothesis[58,59]. Such biases may result from alterations of evidence weighting: giving too strong a weight to initially encountered observations, and to observations that are inconsistent with the current belief. Our task and analytic approach readily captured both forms of cognitive bias in terms of the evidence weighting profiles shown in Fig. 2c–e: jumping to conclusions in the form of a weaker weighting of evidence appearing late compared to early in the sequence (Fig. 2c), as captured by our kernel half difference measure; and bias against disconfirmatory evidence in terms of a reduced up-weighting of the evidence associated with high *CPP* (Fig. 2d), as captured by the overall magnitude of the *CPP* modulation kernels. We expected these changes to be expressed in individuals with increased psychosis proneness.

To evaluate the relationship between psychosis proneness and the detailed shapes of the weighting profiles, we first grouped participants into bins (quintiles; $N = 18$ per bin) defined by the associated P-scores and focused our comparisons on the two participant subgroups with the lowest and highest P-scores (Fig. 3; see also Supplementary Fig. 3). The mean scores were 1.08 (95% C.I. = [1.06, 1.11]) for the 'low P subgroup' and 1.91 (95% C.I. = [1.84, 1.99]) for the 'high P subgroup'. The latter corresponds well to the mean P-score obtained in individuals fulfilling the criteria for ultra-high-risk status for psychosis (1.9, 99% C.I. = 1.71–2.02)[52]. We also computed continuous correlations between scalar indicators derived from the kernels (as well as other measures of behavior) and the individual P-scores from the entire sample (Supplemental Figs. 4 and 5).

There was a clear relation between the P-scores and the model-estimated noise parameter (increased noise for high P-scores; Supplementary Fig. 4d, h), which is consistent with previous observations linking psychosis to similar model parameters estimated from other belief updating

tasks (reviewed in ref. 60); and mixed evidence for an association with subjective hazard rate $\hat{H}$ (Supplementary Fig. 4c, f). Further, participants' overall behavioral performance – evaluated as accuracy, and consistency of choices with those of the ideal observer, specifically on trials containing at least one change point – was better in the low P subgroup than the high P subgroup (Supplementary Fig. 4a, b), albeit without a robust effect for the corresponding continuous correlations (Supplementary Fig. 4e, f). These observations, along with the inferred high-risk status of the well-isolated high P subgroup, motivated our subsequent main focus on the comparison between the two subgroups in our analysis of evidence weighting profiles as well as pupil dynamics.

Importantly, the high P subgroup gave overall less weight to evidence arriving late in the evidence stream, as quantified in terms of the individual sample comparisons (Fig. 3a; sample position 9: mean group difference = 0.45, 95% C.I. = [0.16, 0.74], $P = 0.005$; sample position 10: mean group difference = 1.00, 95% C.I. = [0.29, 1.75], $P = 0.011$; two-tailed permutation tests; all other sample positions not significant) as well as the strength of the above-described overall recency effect (i.e., kernel half difference, Fig. 3b). This same group difference was also significant when normalizing each participant's kernel to have unit area before computing the kernel half difference (Methods; ref. 49) in order to remove any contribution of overall kernel magnitude (mean group difference in normalized kernel half difference = 0.04, 95% C.I. = [0.01, 0.07], $P = 0.013$).

Furthermore, the high P subgroup exhibited a generally weaker dynamic component to their belief updating, reflected in smaller combined weights of the *CPP* and -|ψ| modulations identified above (averaging regression coefficients over largest temporal clusters identified in whole-group analysis, and then summing the averaged coefficients across *CPP* and -|ψ| modulations; Fig. 3c, d). The resulting value, which captured the extent to which individuals implemented the dynamic, non-linear evidence accumulation prescribed by the normative model, was significantly different between low and high P subgroups (Fig. 3d; also marginally significant for *CPP* alone, Supplementary Fig. 5). Similar effects were observed for the kernel half difference, but not for the summed or individual modulation effects, in the continuous correlations across all participants (Supplementary Fig. 5).

We note here that the high P subgroup included one individual with a prior diagnosis of psychosis due to substance abuse (P-score = 2.1). However, with the exception of the combined *CPP* and -|ψ| modulations (Fig. 3d; $P = 0.1$), all effects in Fig. 3, as well as Figs. 4 and 5 below, remained statistically significant when this individual was excluded.

In sum, the above results replicate previous findings with the same decision-making task[17], here in a community sample of participants more representative of the general population than the samples used in previous studies of the computational and neural bases of belief updating in changing environments (but see ref. 22).

**Fig. 5 | Relationship between individual belief updating and pupil metrics.** Scatterplots show across-participant correlation between encoding of CPP and -|ψ| in sample-related pupil responses (pooled) and the two evidence weighting (kernel) metrics from Fig. 3b, d: specifically, variable encoding in pupil response plotted against (**a**) kernel half-difference and (**b**) summed CPP and -|ψ| modulation weights. Circles, participants (*n* = 90); circle colors, low P subgroup (light gray, filled), high P subgroup (black, filled) and remainder of sample (gray, unfilled). Correlation coefficients and *P*-values, Spearman correlation. See Supplementary Fig. 7c, d for the correlations evaluated separately for CPP and -|ψ|.

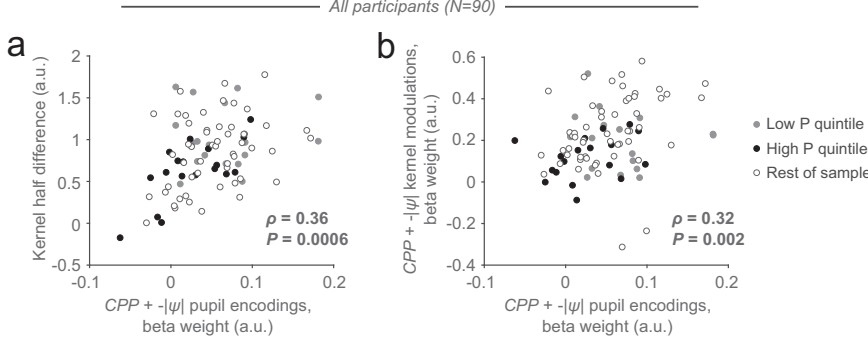

In sum, we found differences in the dynamic, non-linear belief updating process between individuals scoring on the lowest end of the psychosis continuum, and those with scores approximating those of individuals who fulfill established criteria for being at ultra-high risk of developing psychosis[52]. These alterations were in line with a stronger 'stickiness' of belief states formed early on during processing of evidence streams in the high P subgroup.

### Participants prone to psychosis also have altered pupil dynamics

Previous work has shown that phasic arousal encodes *CPP*, uncertainty, and related variables and, at least in part, mediates their impact on learning rate and evidence weighting[17,19,24,61]. Our central prediction was that participants prone to psychosis would also have reduced pupil responses to evidence samples associated with high CPP or occurring under high uncertainty. To test this, we isolated phasic arousal responses to individual samples in terms of the first temporal derivative of the pupil diameter signal measured during task performance[17,62] (Fig. 4a). Pupil diameter is as an established proxy for changes in arousal state[29,63]. We focused on its first derivative in order to increase temporal precision and specificity for noradrenaline release[32]. We then regressed this rapid response component on single-sample *CPP* and -*IψI* estimates in a time-resolved fashion (Methods), without making assumptions about the generator mechanism[64]. This approach revealed robust pupil encoding of both computational variables (Fig. 4d), again replicating previous findings[17].

The evoked pupil responses also differed between the high and low P subgroups (Fig. 4b–d). The overall responses during the trial were reduced in high P individuals, when quantified as either (i) evoked responses in the raw pupil signal and averaged across the trial epoch (Fig. 4b; marginally significant) or (ii) as the response of the pupil derivative (Fig. 4c) evaluated in an early time window that contained the largest excursion of the derivative response (Fig. 4a, dashed line). Furthermore, the encoding of belief updating variables in the pupil derivative was less reliable in the high P subgroup (Fig. 4e; Supplementary Fig. 6).

In sum, not only did the high P subgroup exhibit altered belief updating dynamics, but their task-related recruitment of pupil-linked arousal was weaker, both in terms of the overall evoked response and sensitivity to the computational variables that modulate the belief updating computation.

### Inference-related pupil dilations predict individual evidence weighting profiles

Given that high and low P subgroups differed in both their above-described behavioral signatures of dynamic evidence weighting (i.e., kernel metrics, Fig. 3b, d) and the encoding of computational variables from the inference process in their pupil responses (specifically, *CPP* and uncertainty Fig. 4e), an obvious question is how the individual differences in these variables related to one another. Indeed, we observed robust correlations between the pupil sensitivity to *CPP* and uncertainty, and both evidence weighting signatures: the kernel difference (Fig. 5a) and the (pooled) modulation of evidence weighting by *CPP* and uncertainty (Fig. 5b). Weaker relationships

to the behavioral evidence weighting measures were present for the overall task-evoked pupil response (magnitude or derivative) across the entire trial (Supplementary Fig. 7a, b). Taken together, these results show that the individual level of recruitment of pupil-linked arousal by inference-related variables reflected individual differences in dynamic evidence weighting.

### Role of other symptom dimensions and working memory

Our analyses focused on the P-score of the CAPE questionnaire because of the rationale underlying the study. Indeed, the relationships with belief updating profiles and pupil dynamics reported above were specific to the P-score, with no credible evidence for effects for the other two scores measuring negative (N) and depressive (D) symptoms, respectively (Supplementary Fig. 8). Bayes factors (BF[10]) for these analyses were between 0.65 and 0.33, indicating anecdotal to moderate evidence in favor of the null hypothesis of no effect of N- or D-scores on evidence weighting or pupil measures[65].

Many studies into the pathophysiology of schizophrenia have focused on working memory[66,67], the ability to hold information online for several seconds and manipulate it for the control of behavior[68,69]. Deterioration of working memory is consistently associated with schizophrenia[70], in particular with the negative subdomain[71], and may provide early cues for the development of the disorder[72]. For these reasons, and to test for the specificity of the effects observed in our belief updating task, we also related the CAPE questionnaire scores to participants' performance in a classical delayed match-to-sample working memory task (Methods). We observed no credible evidence for associations between working memory task performance and scores on any of the three scales that comprise the CAPE (Supplementary Fig. 9; Bayes factors indicated anecdotal evidence in support of the null hypothesis of no effect of P-scores, and moderate evidence for the null hypothesis of no effect of N- or D-scores). This suggests that the relationship between psychosis proneness and decision-making established above is driven by individual differences in the dynamic belief updating process, rather than by a possible contribution of working memory integrity to performance of our decision-making task[73].

### Discussion

Our work sheds light on the relationship between the computational and physiological bases of dynamic belief updating and latent psychopathology. We found that a community sample of participants approximated the normative, non-linear belief updating strategy for solving our task. Computational variables entailed in this process, change-point probability and uncertainty, were encoded in their pupil responses during task performance. A subset of individuals who were prone to psychosis gave less weight to evidence arriving late in the decision process as well as less weight to evidence associated with large change-point probability or arriving under low uncertainty. These behavioral signatures are diagnostic of a stronger persistence of initially formed belief states at a computational level. Mechanistically, they are consistent with stronger cortical attractor states, in which sensitivity to new evidence is reduced. These behavioral changes in

psychosis-prone participants were associated with less precise encoding of change-point probability and uncertainty in pupil responses, consistent with our hypothesized involvement of arousal mechanisms in the aberrations of dynamic belief updating. Together, our results integrate different fields of current psychosis research into a single mechanistic framework.

## Assessing the psychosis continuum through questionnaires

Delusions and hallucinations are prominent symptoms of schizophrenia but are also reported by people in the general population in an attenuated form. This has led to the notion of a psychosis continuum[5] ranging from people who report never to have experienced even the mildest type of psychotic experience to those diagnosed with schizophrenia. Located in between these two extremes are various degrees of risk states characterized by subclinical psychotic experiences of varying frequency and intensity. Self-report measures of psychotic experiences, such as the CAPE questionnaire employed here[37], are commonly used to estimate a person's position on this continuum, with higher scores indicating higher proneness to psychosis. The CAPE has been extensively validated[34,37,51,52,74] and shows good test-retest reliability[37]. The self-reported dimensions of psychosis that it purports to assess are associated with the corresponding interview-based dimensions[37,39]. The P-scores that we focused on are suited to screen for psychosis risk[51,52] and for clinical psychosis[74]. The 3-dimensional structure is stable across different populations and language versions[34,75]. We, therefore, assume that the self-reported dimensions of psychotic experiences assessed here are reliable and valid.

## Theoretical foundation of our approach

Our current approach built on recent computational and physiological insights into group average behavior on the same task as used here[17]. This previous work established that (i) participants approximated the normative belief updating strategy for this task, (ii) the underlying belief state was selectively encoded in the slow dynamics of action plans in parietal and frontal cortical regions, (iii) the computation of this belief state emerged from recurrent interactions in local microcircuits equipped with attractor dynamics, and (iv) pupil-linked phasic arousal responses contributed to the dynamic upweighting of evidence samples associated with high change-point probability and uncertainty, a process that was linked to modulation of the evidence encoding in visual cortex.

This prior work also (v) identified diagnostic signatures of the non-linear, dynamic belief updating process in the form of evidence weighting profiles (psychophysical kernels)[17], which formed the cornerstones of our current behavioral approach. Our findings highlight the importance of the detailed, model-guided quantification of evidence weighting profiles widely used in decision neuroscience[17,76] – in particular, the diagnostic CPP and uncertainty modulation kernels that our previous work derived computationally and validated physiologically[17]. These profiles can identify even nuanced and idiosyncratic alterations in the dynamic weighting profiles (see also ref. 77 for an application relating hallucination proneness to weighting of different stimulus features during auditory decision-making), which may not be fully captured by parameter estimates from model fits. Consistent with some previous work (reviewed in ref. 60), we found an effect of individual P-scores on decision noise and a weak effect on the subjective hazard rate (dependent on inclusion or exclusion of an outlier). But this combination of parameter effects only partially captured patterns observed in our kernel-derived measures (underestimating particularly the modulation of evidence weighting by CPP). In line with recent insights from a learning task with change-points applied to schizophrenia[78], this pattern of results indicates that static parameter estimates can fail to capture important deviations in belief updating dynamics. Such deviations are instead well captured by the psychophysical kernels that we here estimated for each individual.

## Identifying reasoning biases in psychosis

Our findings are broadly consistent with accounts of aberrant probabilistic reasoning in psychosis that focus on the over-expression of two cognitive biases: jumping to conclusions[10,59] and bias against disconfirmatory evidence[58,59]. As such, the findings align with cognitive models of delusions that posit these reasoning biases to be a central part of psychotic vulnerability and to account for delusional interpretations of ambiguous stimuli[79–83]. However, existing accounts of biased probabilistic reasoning are largely based on results from the so-called beads task, in which the participant sequentially draws colored beads from one of two jars associated with different probabilities of one versus the other color, and has to judge which jar they are drawing from[84,85]. This task, and the associated behavioral analyses, suffer from interpretational limitations. First, the neural bases of the cognitive computations underlying performance of these tasks is not well understood. Second, inferences about evidence accumulation (and biases thereof) are indirect, through latent model variables or substitute measures of behavioral performance; as such, the dynamics of belief updating are not probed as directly as through analysis of psychophysical kernels[84,85]. Third, in the classical 'draws to decision' version of the beads task[84,85], the participant draws beads from one of two jars with fixed (and already learned) probabilities until committing to a decision about which of the two jars they are drawing from. This version requires the setting of decision bounds to determine how much evidence is accumulated before the participant decides to stop accumulating and make a response. This may conflate biases in evidence accumulation *per se* with a general proneness to commit to decisions and/or to stop engaging in the task. Fourth, and most fundamentally, the standard tasks that have been used in this literature have lacked hidden state changes, which are an essential feature of natural environments[1]. Indeed, the presence of such hidden state changes is critical for unraveling non-linearities of human belief updating[17].

Our current approach overcomes the above limitations. We used an evidence accumulation task with hidden change-points, which provided direct access to the dynamics of evidence weighting and the neurophysiological basis of which we have previously characterized in detail. This enabled us to precisely quantify two biases through metrics derived from participants' psychophysical kernels: (i) Primacy vs. recency in temporal evidence weighting, captured here by the 'kernel half difference'; and (ii) modulation of evidence weighting by the contextual variables change-point probability (which depends on the inconsistency of a new evidence sample with the current belief state) and, to a weaker extent, uncertainty, captured here as the mean across additional modulatory kernels estimated as part of our analytic approach. Because the number of samples presented on each trial are under the experimenter's (rather than the participant's) control, these biases are independent of stopping rules and directly inform about belief updating. Stronger tendency toward primacy in evidence weighting can be interpreted as a stronger tendency toward jumping to conclusions[86]. Likewise, a reduced modulation of evidence weighting by CPP indicates a bias against disconfirmatory evidence. Both effects were present (the latter trending when isolated from the uncertainty modulation, which is not clearly related to the two biases in question) in participants with the highest proneness to psychosis. Finally, and critically, our approach enabled linking the behavioral identification of these biases to specific, inference-driven components of task-evoked pupil responses.

Importantly, we observed some evidence that the relationship between psychosis proneness and biases in dynamic belief updating was a specific effect. Our evidence accumulation task required working memory (for the maintenance of accumulated evidence)[73,87], and schizophrenia is associated with degraded working memory[70–72,88]. Yet, we here found anecdotal evidence[65] that P-scores were not related to performance on a standard (delayed match-to-sample) working memory task. Furthermore, the 3-dimensional structure of the CAPE allowed us to assess whether signatures of dynamic belief updating were related not just to the positive symptoms of psychosis proneness captured by the P-scores, but to self-reported levels of negative and depressive symptoms. Again, we observed anecdotal to moderate evidence against the presence of such relationships. Thus, the relationship between biases in dynamic belief updating and positive symptoms of psychosis proneness could not be explained by either individual differences in working memory integrity, or general effects that also manifest in other symptom dimensions.

## Outlines of a mechanistic framework

Our results open a window on the pathomechanisms of psychosis. We observed that the precision of the encoding of CPP and uncertainty in pupil responses, a high-level, cognitive component of the pupil dynamics in our task, was particularly closely related to individual evidence weighting signatures and was also related to delusion proneness. Pupil diameter is as an established proxy for arousal state[29,63], and reflects the activity of ascending neuromodulatory systems such as the locus coeruleus-noradrenergic system[27-32]. It is tempting to speculate that – in keeping with the opponent interplay between tonic and phasic activity of central arousal systems[25] – the previously established tonic hyper-arousal in psychosis[12-16] blunts evoked, phasic arousal responses to high-level variables computed in the process of belief updating, specifically CPP and uncertainty. This in turn reduces the dynamic upweighting of internal representations of the momentary evidence observed in sensory cortex[17], translating into a smaller than required update of the belief state in downstream cortical regions. As a consequence, an individual's beliefs will become 'sticky'. Our findings were obtained in an emotionally neutral belief updating task operating on short timescales and unrelated to the common (often social) contents of psychotic beliefs. It is all the more striking that specific computational measures of behavior and arousal response derived from this kind of task context relate to overall psychosis proneness. This suggests that the pathophysiology of psychosis may affect not only high-level (social, emotional) reasoning that dominates delusions, but also inferences about emotionally neutral properties of the sensory environment.

## Relationship to predictive coding accounts of psychosis

It is instructive to relate our results to predictive coding accounts of psychosis[4,89-91]. Our results are broadly consistent with certain assumptions of these accounts. The reduced pupil encoding of the inconsistency between prior and evidence (as measured by change-point probability) that we observed here in the high-P subgroup suggests an impaired computation of high-level (cognitive) prediction errors, which other work has linked to schizophrenia and delusion severity[92]. Furthermore, it is tempting to relate our observation of stronger stickiness of initially formed beliefs to the notion of overly strong priors about a higher-level context as a source of false beliefs[93]. However, previous psychophysics work points to weak lower-level (perceptual) priors and/or more precise sensory likelihoods in psychosis, manifesting in a lower sensitivity to some visual illusions[94,95]. While our task can be described as a visual evidence accumulation task, it is cognitive in nature in the sense that all evidence samples are clearly discernible (clearly above detection threshold) and it requires a highly flexible formation and updating of prior beliefs, different from many perceptual illusions that tap into hardwired sensory priors. This may explain why our results clearly favor accounts of psychosis that emphasize strong priors[94]. Here, we put the focus on the dynamic evolution of belief states during the accumulation of sequentially presented sensory evidence. This enabled us to identify the dynamic interplay between beliefs and evidence as a key aberration in psychosis-proneness.

## Limitations

Our work also has limitations. On the one hand, we sampled participants entirely from the general population without current diagnoses of psychosis or related clinical conditions; our multi-session, in-laboratory data collection meant that our sample size was limited, particularly relative to similar work leveraging online data collection (e.g., ref. 96.); and the behavioral, model-based and pupil differences that we observed even between extremes of the measured part of the P-score continuum were generally small. Future work should assess the degree to which the current findings generalize to individuals with stronger symptoms (e.g., diagnosed ultra-high risk status or full diagnosis of schizophrenia), ideally with a larger sample – and supplement the current approach with direct assessment of baseline arousal levels[13,14] to test the hypothesis that tonic hyper-arousal is a mediator of the effects reported here. On the other hand, our strategy for sampling participants involved slight oversampling in the higher half of the P-score distribution (Methods). It is therefore possible that the current sample was biased towards psychopathology, and our findings should be interpreted

with this in mind. We emphasize that the sole purpose of our oversampling was to counteract differences between the sample that emerged during data collection and the distribution of P-scores from a larger, and therefore more representative, community sample[34]. It is, therefore, unlikely that our sampling approach introduced a bias toward higher P-scores, or psychopathology more generally. In keeping with this conclusion, the levels of self-reported previous diagnosis of mental health issues in our sample are at the low end of the rates typically found in the general population (e.g., only 4.4% of our participants reported a previous diagnosis of depression).

## Conclusions

We found that aberrations in belief updating and arousal dynamics are evident in individuals who have moderate-to-high levels of self-reported psychosis proneness but have not been clinically diagnosed with schizophrenia. We speculate that these aberrations may give rise to psychotic symptoms (i.e., delusions), by increasing the tendency to hold on to erroneous beliefs. They may also serve as a risk marker of psychotic psychopathology. Our computational approach to understanding belief updating behavior and arousal may become a useful tool for future studies into the pathophysiology of psychosis and other mental disorders.

## Data availability

Raw behavioral, eye-tracking and questionnaire data used to generate our results, as well as numerical data underlying all main text and Supplementary Figs., are available at the open access repository linked to in ref. 97 (https://doi.org/10.25592/uhhfdm.14759).

## Code availability

Analysis code is available without restriction at https://github.com/murphyp7/2024_Murphy_Belief-updating-psychosis-proneness. All code has been tested using Matlab version 2021a.

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

## Acknowledgements

We thank Emma Krink, Marlene Petersson, and Sonja Monien for strong engagement in data collection. This work was funded by the Deutsche Forschungsgemeinschaft (DFG, German Research Foundation) projects DO 1240/4-1, and SFB 936 - Projekt-Nr. A7 (all to THD). The funders had no role

in study design, data collection and analysis, decision to publish or preparation of the manuscript.

## Author contributions

P.R.M.: conceptualization, methodology, software, formal analysis, visualization, writing – original draft, writing – review and editing; K.K.: conceptualization, methodology, software, writing – review and editing; G.M.: investigation, formal analysis, writing – review and editing; N.K.: investigation; T.L.: conceptualization, methodology, resources, writing – review and editing, supervision; T.H.D.: conceptualization, methodology, resources, writing—original draft, writing—review and editing, supervision.

## Funding

## Competing interests

The authors declare no competing interests.
