## [Peer Review File · Communications Psychology]

26th Mar 24

Dear Dr Murphy,

Thank you for your patience during the peer-review process. Your manuscript titled "Individual Differences in Belief Updating and Phasic Arousal Are Related to Psychosis Proneness" has now been seen by 3 reviewers, and I include their comments at the end of this message. They find your work of interest but raised some important points. We are interested in the possibility of publishing your study in Communications Psychology, but would like to consider your responses to these concerns and assess a revised manuscript before we make a final decision on publication.

We therefore invite you to revise and resubmit your manuscript, along with a point-by-point response to the reviewers. Please highlight all changes in the manuscript text file.

Reviewers agree that the experimental data and computational modeling are solid, however they also agree that the results are not clearly communicated. Therefore, editorially we would like you to incorporate the suggestions of the reviewers that might help refocusing the introduction and the discussion around an effective main message. This message should be easily understandable to a broader audience. The idea of integrating the literature on arousal and belief updating in psychosis is innovative and bridges findings from different fields in one single mechanistic explanation, we would like to see this message more effectively communicated. The results section also needs to be more clearly explained, the three reviewers who are all experts in the field highlighted several difficulties in following this section, reviewer#1 and reviewer#2 have some useful suggestions about this aspects. Finally, we would also like you to address the concern of reviewer #3 about the potential contradiction between the results from the noise parameter estimation and the recency effect.

I am attaching an Editorial Requests Table that details critical reporting requirements for the revised manuscript. Please attend to each item and ensure your manuscript is fully compliant. We are requesting that your manuscript aligns with these requirements as this facilitates the evaluation of your manuscript, reducing delays in re-review and potential future acceptance. If your revised manuscript is not aligned with these requests on major issues, such as those concerning statistics, it may be returned to you for further revisions without re-review. Additional information can be found in our style and formatting guide Communications Psychology formatting guide.

Please use the following link to submit your

- revised manuscript,
- point-by-point response to the referees' comments,
- cover letter (as a separate document),
- the Editorial Policy Checklist (see below),
- the Reporting Summary (see below), and
- the completed Editorial Request Table (attached):

[link redacted]

Best regards,

Eva R. Pool

Eva R. Pool, PhD

Editorial Board Member

Communications Psychology

orcid.org/0000-0001-5929-1007

REVIEWER EXPERTISE:

Reviewer #1: Belief updating and Psychosis

Reviewer #2: Computational Psychiatry and Schizophrenia

Reviewer #2: Computational Psychiatry and Psychosis

REVIEWER REPORTS:

Reviewer #1 (Remarks to the Author):

SUMMARY: Thank you for the opportunity to review this article. This study by Murphy et al. investigates how people's ability to update their beliefs based on new information relates to their likelihood of experiencing psychosis-like symptoms. They explore this by having participants with subclinical psychotic experiences (i.e., a community sample) perform a task where they need to make decisions based on changing cues, while also measuring how their pupils change size – an indicator of arousal and attention. They found that people who are more prone to psychosis, as measured by the CAPE questionnaire, tend to stick too much to their initial beliefs and don't adjust them flexibly with new evidence. These participants also showed less change in pupil size, suggesting differences in how their brain processes arousal and adapt to new information.

Major concern:

1. Experimental

A. Methodological details and task validation. The main behavioral task is a two-alternative forced-choice task involving hidden state changes and evidence accumulation, designed to mimic naturalistic decision-making environments.

a. Concern: However, the authors do not seem to provide detailed justification for the specific task parameters chosen (e.g., hazard rate of 0.1, sequence lengths, and evidence sample distributions).

b. Suggestion: Maybe outlining a table as such in the Supplementary would be helpful. It would make the study's design more transparent to readers. This can facilitate replication. Providing this type of rationale also shows how each aspect of the task is intended to probe specific cognitive processes.

Task parameter Description Chosen value Justification

Hazard Rate (H) Prob. of state change per trial 0.1 Selected to balance between too frequent or too infrequent changes

2. Interpretational

A. The last sentence of the abstract holds a general claim that might be overstating the study's immediate implications. I suggest changing it to something like "Our findings provide new insights into the cognitive and physiological processes associated with psychosis proneness, offering potential pathways for future research into the mechanisms underlying psychotic disorders."

Minor concern:

1. Experimental

B. Sample selection and diversity. The study's sampling strategy aimed to oversample individuals with higher psychosis proneness scores, aiming for 50% of participants to have a P-score above the 50th percentile within a large community sample.

a. Concern: While this approach is beneficial for studying psychosis proneness, it may introduce biases related to the overrepresentation of certain psychological profiles, potentially affecting the generalizability of the findings.

b. Suggestion: It seems like a prior work studied normative belief-updating in the general population. Nonetheless, clarifying how this oversampling strategy impacts the study's conclusions in the Discussion could help with transparency.

2. Analytical

A. Psychophysical kernel analysis. The use of psychophysical kernels to quantify the impact of evidence on choice over time provides rich insights into the dynamics of belief-updating.

a. Concern: However, the interpretation of these kernels, particularly in terms of the relative weighting of early versus late evidence and its modulation by CPP and uncertainty, would benefit from a more in-depth discussion.

b. Suggestion: Explaining how these findings align with or diverge from existing theories of cognitive biases in psychosis would add depth to the discussion.

B. Pupil dynamics. The analysis of pupil dynamics and its relation to cognitive variables such as CPP and uncertainty is a novel aspect of the study.

a. Concern/Suggestion: While the findings suggest altered arousal dynamics in individuals with high psychosis proneness, a deeper exploration of the neurophysiological underpinnings of these differences would enrich the discussion. Additionally, considering the role of tonic arousal levels and its interaction with phasic responses could provide a more comprehensive picture of arousal regulation in psychosis proneness.

3. Figures

A. Figure 5 legend. Given that the authors are studying individual belief-updating differences between low and high psychosis proneness, it would be good to see a legend for low and high P quintile; that is, color-code the individual data points.

OVERALL EVALUATION: The study presents a novel approach to understanding the mechanisms of psychosis proneness, offering valuable insights into the interplay between belief-updating, phasic arousal, and psychosis. Its integration of behavioral and physiological measures provides a comprehensive analysis in consilience with their prior work using MEG will pave way for future research in psychopathology. The methodological rigor is commendable, with a well-structured experimental design and robust statistical analysis. The innovative use of pupillometry to measure cognitive and physiological responses adds depth to the findings. While the study is thorough, certain limitations in sample diversity and the potential for broader generalization of results should be addressed. Additionally, further clarification of (i) task parameter selection and (ii) general claims about the results might strengthen the paper. Given the paper's substantial contributions and methodological strengths, with consideration of its few weaknesses, this is impactful work. Addressing the points mentioned would enhance the clarity and impact of the findings.

Reviewer #2 (Remarks to the Author):

Overview

- The authors present a paper that is methodologically very interesting. Use of pupillometry in conjunction with computational modelling to task data – and linking to psychosis – is exciting. However, after careful deliberation, I would recommend rejection and that the authors work on several key points before resubmitting. Namely – 1. structure – intro and discussion need to refocus on expressing the key ideas, gaps and context of the work. Think about relevance and coherence in the writing – alongside traditional research paper structure; 2. Analyses – a lot of great work here clearly, but again – its not communicated well – its not clear why you chose the model/s you did for example – as a reader I shouldn't need copious amounts of domain knowledge or to do to another paper to get background info – which I felt as though I did here. Similarly - how the analyses relate to specific pre determined hypotheses, this is missing – and alongside some slightly odd reporting of sensitivity analyses, raises question marks. 3. Importance and implementation – why is this research vital to field, how does it relate to psychosis – what does it mean for someone with psychosis? How does your highly specific question link back up to the wider world?

General thoughts by section

Intro

- The link between psychosis symptoms and belief updating under uncertainty could be improved – the paper discusses previous computational modelling findings in relation to tonic/phasic differences, but it is not immediately clear to a reader e.g., how these differences may explain symptoms in psychosis (like delusions) – what does it mean to say that there is an increase of tonic and/or phasic arousal levels? (unnecessary belief updating/ perceive the world as volatile)
- Mention of 'adaptive' belief updating – but not clearly defined how the models/tasks that you use will get at adaptive aspects of belief updating – other references that could be useful are those from Mike Browning's lab at Oxford., – integrates adaptation to volatility and pupillometry (although is in anxiety)
- Some further elaboration of tasks and how they capture aspects of information processing could be useful
- Uncertainty in this context is described only in relation to volatility. The first paragraph indicates “multiple sources of uncertainty” but this is not expanded upon or stated that the paper will only focus on volatility – links to the description of the normative account of learning (e.g., Piray & Daw, 2021a; pulcu and browning 2021)

- The intro talks about jumping to conclusion tasks, but while the task used here is similar, the analysis isn't looking at accumulation of evidence (using drift diffusion methods e.g.) but the individuals assessment of volatility. This might be useful info in the intro
- In general I feel the intro failed to explain and motivate the research - and importantly the gap in existing literature that this research is filling

Methods

- It was unclear to me how the "oversampling" approach was done. From the information given it could not be directly replicated - e.g., how many additional people did you sample to get to the amount needed, or are the authors talking about bootstrapping? - how many people ended up in each group? It says "aiming for 50% with a Pscore above 50th percentile - does that mean that in the subsamples there were n=45 in low and n=45 in high?"
- The differences between "symptom" groups were still very small - feature of general population sample - a larger sample would be better (not a helpful comment I realize but perhaps can be reflected in the conclusions that are drawn from the research/future directions)
- Possibly the inclusion of the individual with psychosis in figure 3d results is misleading - I would suggest reporting in the results section too
- Computational Modelling - a normative model was compared with a perfect match (ideal observer) model, and final evidence models i.e., recency. Model comparison (relative goodness of fit) was computed using BIC - where BIC including the n free parameters and trials
 - o It says that 4 different model variants were fit -but I am unclear what these were exactly - are the results of the model comparisons reported? It directs the reader to supplementary fig 2 - but this doesn't clear things up for me
 - o The winning normative model includes 2 free parameters - the process of getting to this model configuration could be more clearly explained
- Recency/primacy effect - "psychophysical kernels" using logistic regression - and then computing the "kernel half difference".
 - o The 2 sample t test used to compare groups not fully described -any covariates? Not enough information about the analysis here.

Results

Psychosis group agnostic results

- Behavioral analyses showed that all participants performed within expected limits but were close to ideal observer than other strategies. However - these results should be more clearly presented

- At an individual level its suggested there was significant variation – but I am unclear about how this is tested or what the word significant refers to

- The figure 2 legend could be improved upon – esp. from d onwards.

- The authors calculate the change point probability and the “uncertainty” using the normative model – and compute a modulation analysis that uses regression weights to determine the temporal integration of evidence – i.e., recency and primacy effects – importantly comparing the first and last 5 trials. It would be useful to more clearly explain how this is modulation and/or adaptation when its only the first and last 5 trials being compared

- The summary of findings in section “participants implement adaptive belief updating with non linear evidence weighting” is unclear

Results pertaining to psychosis symptoms

- “Grouped kernels into bins (quintiles) defined by the associated p-scores” - I think authors are saying they took a subset of the sample (highest and lowest) but it doesn’t say how many people this is – and how it relates to kernels...

- First authors report that change point trials – proportion correct – differs significantly between groups – supp material fig 4a shows this but doesn’t report effect sizes

o Also they are reporting both the comparison between groups (quintiles) and the correlation. Computing both analyses needs to be both better justified, and also should be clearly related to pre defined hypotheses clearly stated up top.

- the noise parameter differed between groups – supplementary fig 4d and h – decision noise increased in high group – relates to previous findings for decision noise (temperature parameters) in psychosis refs – experiment - Katthagen et al., 2018, review - Gibbs-Dean et al., 2023

- subjective hazard rate – The way its written seems to suggest that the sensitivity analyses removing the individual with psychosis disorder is selectively applied to some analyses and not others. If this is not the case it needs to be better explained. If it is, this is problematic if not clearly justified beforehand (and perhaps even then)

- Findings for the kernels and psychosis:

o So here authors talk about adaptive component to belief updating –Its described as the summed regression coefficients across change point, and uncertainty modulations. I was lost in this discussion, its is not that complicated I think but investment needed in clearer explanations

Discussion

- Unusual to start the first paragraph of the discussion solely with what previous findings have shown. I would be better to clear state your findings in the first paragraph of the discussion,

o – speaks to a wider problem about clearly stating the aims, hypotheses, and crucially the importance and implementation possibilities for this research – If the hypotheses were more clearly defined then it would be easier to relate the findings back

- Lack of discussion about (lack of) findings for negative and depression. Similarly, where some findings are in line with wider literature e.g., psychosis and decision noise

- The findings aren't very clearly contextualized in relation to previous literature – after stating the findings clearly – how do this relate to other research e.g., linked back to psychosis literature and symptoms in details

- The findings that are discussed are overstated for what (I think) the statistical results actually show.

- Does the paper represent an advance in understanding which may influence thinking in the field?

The authors suggest that they have replicated findings that have been shown in psychosis groups, in a “more general” population. It's not clear what the importance of this is – I think authors could improve on clearly stating the importance of the research, and how it further advances the field

- Does the article present an original study, new analysis, new model, or a direct or extended replication of previous work?

Yes - While the research is partially aiming to replicate previous work, there is originality in the methods – e.g., pupillometry + computational modelling in psychosis.

- Are the data and analysis technically sound? Are they appropriate to answer the research question, e.g., are causal research questions addressed on the basis of causal, rather than correlational evidence?

I found it hard to tell, as I felt the methods and analysis were not very clearly explained. The analysis was not clearly hypothesis driven, which affected my ability to assess it technically

Does the paper provide strong evidence for its conclusions?

I think here improvements could be made. The sample is small, effect sizes appear small. The conclusions drawn are quite strong for what the results actually are, perhaps overstated somewhat.

-Is the study question important to scientists for a sub-field of psychology?

I believe it is, but I think the authors need to think about how to better communicate this to the audience

Are there any special ethical concerns arising from the use of animals or human subjects? No

- Was the study preregistered and if so, did the authors follow the preregistration? Not preregistered as far as I can tell

Reviewer #3 (Remarks to the Author):

Summary:

This study investigated the relationship between evidence accumulation and pupil dilation during a volatile belief updating task (i.e., with change points). Their rationale for the study was that previously identified increased tonic arousal in psychotic populations would likely drive reduced phasic arousal, which in turn may impact pupil responses. Because of the altered pupil responses patients may have altered evidence accumulation. The purpose of this study is to assess the link between phasic arousal, as measured by pupil responses, and evidence accumulation processes, as measured by behavioral and computational metrics. The study was conducted in healthy controls in line with the psychosis continuum framework.

The authors replicated previous results in this larger sample, finding that participants update their beliefs in this task consistent with the normative model presented in the paper. The model clearly fits the data well and best fit parameters can replicate participant behavior. The supplement contains model validation and comparison measures, consistent with best practices. There is also a good degree of interindividual variability across fitted parameters. The authors note in the

methods that their recruitment strategy enriched the sample for higher P-score participants, which is often necessary for this kind of work to obtain the needed range of scores to test for interindividual differences.

To assess the effect of psychosis proneness, participants were grouped into quintiles by P-score and focused on group-comparisons of the highest and the lowest P-groups, but dimensional correlations in the supplement including all participants were consistent with these results. The primary findings were a positive relationship between P-score and the noise parameter across all participants (i.e. more noise associated with more psychosis proneness), general performance differences across the high and low P-group, and differences in belief updating dynamics.

In terms of pupil dynamics, the authors show weaker pupil-linked arousal for the high P group and that the degree of encoding is associated with both behavioral and computational measures of belief dynamics (Kernal Half Difference/ CPP+Phi kernal) across all participants.

This is a very nice paper. The task is well designed, the computational modelling is top-notch, and the pupillometry provides a novel contribution to the literature. I have a few comments that I hope will help the authors improve the manuscript. Great job!!

Introduction

Page 2: “The normative process entails a dynamic ‘upweighting’ of new evidence at moments when the probability of a state change (‘change-point probability’, CPP) or uncertainty in the belief are high.” It has also been suggested that there may be down-weighting of prior knowledge, which may produce similar results. For example: <https://www.ncbi.nlm.nih.gov/pmc/articles/PMC8041039/>

The authors may want to briefly address this distinction and why their study/intro focuses on upweighting of new information.

Page 2: “Thus, increased tonic arousal levels observed in psychosis may reduce the phasic, pupil-linked arousal responses tracking CPP or uncertainty during belief updating. This, in turn, may reduce the dynamic evidence reweighting in changing environments, altering the belief updating process overall.” I think it may be useful to clarify this sentence a little more and connect for the reader more why differences in pupil-linked arousal may reduce dynamic reweighting. I can see how this might work in a visual context, but since it doesn’t extend to other sensory domains, and considering that most hallucinations are auditory, I think some more context would help.

Results

Page 6: “While the previously discovered evidence weighting effects were statistically significant at the group level, we also observed substantial individual differences in the evidence weighting profiles (Figure 2c-e).” Could you clarify the “Group” here, in “group-level”? Are you referring to the whole group, or the P-score groups?

Page 6 – 8: My only major question is related to the relationship between psychosis proneness and the computational parameters. The authors report a positive relationship between P-score and the noise parameter across all participants (i.e. more noise associated with more psychosis proneness) in Supplemental figure 4h. Given this, I was surprised to see that the high P group showed more primacy/less recency than the low P group. I would have expected that the noisier group would have more recency, since prior beliefs would be less reliable leading to more reliance on newer evidence. Some previous work has shown this and is at least consistent with the noise results you found (<https://pubmed.ncbi.nlm.nih.gov/36548395/>). In figure 2a the high P group (i.e. the noisy group) appears to update slower than the low P group, also consistent with the notion of unreliable priors (though it’s possible I am misunderstanding the y-axis). Could you speak to this seeming contradiction?

Also, more generally, could you plot along with the data the best fit model predictions (i.e. a posterior predictive check) for figure 3. It’s possible that I am just mis-modeling the task in my head and these model checks would clarify what is “supposed” to happen here.

EDITORIAL POLICIES

We ask that you ensure your manuscript complies with our editorial policies and reporting requirements.

To that end, we require revised manuscripts to be accompanied by two completed items: a reporting summary that collects information on study design and procedure, and an editorial policy checklist that verifies compliance with all required editorial policies.

Nature Research Reporting Summary

Editorial Policy Checklist

All points on the policy checklist must be addressed. Your revised manuscript can only be sent back to the referees if these checklists are completed and uploaded with the revision.

Notes: If you have submitted a Stage 1 Registered Report, Review, Primer, Comment, or Perspective you do not need to submit these forms. If you have already submitted these forms, you may disregard this request.

COMMSPSYCHOL-24-0043-T: Response to Reviewers

We sincerely thank all reviewers for their constructive evaluations of our manuscript. In what follows, we provide a point-by-point reply to all reviewer comments (original comments presented in blue font, our replies in black).

Reviewer #1 (Expertise: Belief updating and Psychosis):

SUMMARY: Thank you for the opportunity to review this article. This study by Murphy et al. investigates how people's ability to update their beliefs based on new information relates to their likelihood of experiencing psychosis-like symptoms. They explore this by having participants with subclinical psychotic experiences (i.e., a community sample) perform a task where they need to make decisions based on changing cues, while also measuring how their pupils change size – an indicator of arousal and attention. They found that people who are more prone to psychosis, as measured by the CAPE questionnaire, tend to stick too much to their initial beliefs and don't adjust them flexibly with new evidence. These participants also showed less change in pupil size, suggesting differences in how their brain processes arousal and adapt to new information.

Thank you very much for your assessment of our manuscript and your helpful suggestions, which we have now addressed.

Major concern:

1. Experimental

A. Methodological details and task validation. The main behavioral task is a two-alternative forced-choice task involving hidden state changes and evidence accumulation, designed to mimic naturalistic decision-making environments.

a. Concern: However, the authors do not seem to provide detailed justification for the specific task parameters chosen (e.g., hazard rate of 0.1, sequence lengths, and evidence sample distributions).

b. Suggestion: Maybe outlining a table as such in the Supplementary would be helpful. It would make the study's design more transparent to readers. This can facilitate replication. Providing this type of rationale also shows how each aspect of the task is intended to probe specific cognitive processes.

Task parameter Description Chosen value Justification

Hazard Rate (H) Prob. of state change per trial 0.1 Selected to balance between too frequent or too infrequent changes

We have now clarified these points in the manuscript, through both additional detail in the Methods section (p.5) and, as you suggest, a table providing additional description and motivation for each task parameter (Supplementary Table 1).

In brief, our generative statistics and sequence lengths were chosen to produce highly expressive belief updating dynamics, striking a balance between allowing strong beliefs to be formed and the presence of regular change-points to produce periods of high surprise and uncertainty. Our choice of H and generative signal-to-noise ratio was based on previous simulations of the normative model reported in Murphy et al. (2021; Extended Data Fig. 1): this parameter combination yielded strong modulations of evidence weighting by both CPP

and uncertainty. Those modulations, in turn, are the diagnostic features that set the normative (non-linear) model apart from simpler schemes of belief updating such as perfect or leaky evidence accumulation.

2. Interpretational

A. The last sentence of the abstract holds a general claim that might be overstating the study's immediate implications. I suggest changing it to something like "Our findings provide new insights into the cognitive and physiological processes associated with psychosis proneness, offering potential pathways for future research into the mechanisms underlying psychotic disorders."

We agree that the final sentence of our original abstract may have overstated the implications of our findings. We have now modified this statement along the lines of your suggestion.

Minor concern:

1. Experimental

B. Sample selection and diversity. The study's sampling strategy aimed to oversample individuals with higher psychosis proneness scores, aiming for 50% of participants to have a P-score above the 50th percentile within a large community sample.

a. Concern: While this approach is beneficial for studying psychosis proneness, it may introduce biases related to the overrepresentation of certain psychological profiles, potentially affecting the generalizability of the findings.

b. Suggestion: It seems like a prior work studied normative belief-updating in the general population. Nonetheless, clarifying how this oversampling strategy impacts the study's conclusions in the Discussion could help with transparency.

Our oversampling in the higher half of the P-score distribution specifically served to rectify differences between our sample and that of a previously reported, larger-scale community sample (Schlier, Jaya, Moritz, & Lincoln, 2015). Therefore, we do not think it likely that our oversampling approach will have biased the sample to the more clinical end. This conclusion is supported by the generally low levels of self-reported previous diagnosis of mental health issues in our sample (e.g. 4.4% with previous diagnosis of depression, compared to 5.2% nationally in Germany).

We now acknowledge in the Discussion section (p.22) that bias due to our sampling approach is a possibility and provide more detail on the specific nature of our sampling strategy in the Methods section (p.3). We realised that the latter was lacking from our initial submission, and we are grateful for you prompting us to look into this.

2. Analytical

A. Psychophysical kernel analysis. The use of psychophysical kernels to quantify the impact of evidence on choice over time provides rich insights into the dynamics of belief-updating.

a. Concern: However, the interpretation of these kernels, particularly in terms of the relative weighting of early versus late evidence and its modulation by CPP and uncertainty, would benefit from a more in-depth discussion.

b. Suggestion: Explaining how these findings align with or diverge from existing theories of cognitive biases in psychosis would add depth to the discussion.

In the Discussion, we now draw more explicit links to theories of delusions that posit cognitive biases (i.e. jumping-to-conclusions bias, bias against disconfirmatory evidence) to be a central part of psychotic vulnerability and to account for delusional interpretations. We also now draw more explicit links between these two forms of cognitive bias and two specific properties of our psychophysical kernels: the difference in weighting given to early versus late samples, and the overall modulation of evidence weighting by CPP, respectively. We do so already in the Results section (p.14) before we describe any results relating the kernel measures to P-scores; and by also making explicit reference to the kernel metrics when linking our findings to the two biases in question in the Discussion section (p.20). We also now make explicit reference to the alignment of the findings in question with cognitive models of delusions and psychotic vulnerability in which reasoning biases play a central part (p.20). Thank you for prompting us to clarify these links.

B. Pupil dynamics. The analysis of pupil dynamics and its relation to cognitive variables such as CPP and uncertainty is a novel aspect of the study.

a. Concern/Suggestion: While the findings suggest altered arousal dynamics in individuals with high psychosis proneness, a deeper exploration of the neurophysiological underpinnings of these differences would enrich the discussion. Additionally, considering the role of tonic arousal levels and its interaction with phasic responses could provide a more comprehensive picture of arousal regulation in psychosis proneness.

In the Introduction (p.2) and Discussion (p.21) sections, we now include additional text that links our approach and pupil results to possible neurophysiological underpinnings (specifically, neuromodulatory systems linked to changes in pupil size), as well as the opponent interplay between tonic and phasic arousal. Please note that, because pupil diameter has been linked in other work with activity of several different neuromodulatory systems, and key experiments have not yet been carried out to differentiate their contributions to pupil fluctuations, we remain cautious about implicating any one system specifically.

3. Figures

A. Figure 5 legend. Given that the authors are studying individual belief-updating differences between low and high psychosis proneness, it would be good to see a legend for low and high P quintile; that is, color-code the individual data points.

We now differentiate between the high and low P-score subgroups and the remainder of the sample in Figure 5 (as well as the related Supplementary Figure 7) by color-coding individual data points, thank you for this good suggestion.

OVERALL EVALUATION: The study presents a novel approach to understanding the mechanisms of psychosis proneness, offering valuable insights into the interplay between belief-updating, phasic arousal, and psychosis. Its integration of behavioral and physiological measures provides a comprehensive analysis in consonance with their prior work using MEG will pave way for future research in psychopathology. The methodological rigor is commendable, with a well-structured experimental design and robust statistical analysis. The innovative use of pupillometry to measure cognitive and physiological responses adds depth to the findings. While the study is thorough, certain limitations in sample diversity and the potential for broader generalization of results should be addressed. Additionally, further clarification of (i) task parameter selection and (ii) general claims about the results might strengthen the paper. Given the paper's substantial contributions and methodological strengths, with consideration of its few weaknesses, this is impactful work. Addressing the points mentioned would enhance the clarity and impact of the findings.

Thank you very much for your assessment of our manuscript and your excellent suggestions. We are very happy that you appreciate our approach and findings and hope you will feel that your concerns have been fully addressed.

Reviewer #2 (Expertise: Computational Psychiatry and Schizophrenia):

Overview

- The authors present a paper that is methodologically very interesting. Use of pupillometry in conjunction with computational modelling to task data – and linking to psychosis – is exciting. However, after careful deliberation, I would recommend rejection and that the authors work on several key points before resubmitting. Namely – 1. structure – intro and discussion need to refocus on expressing the key ideas, gaps and context of the work. Think about relevance and coherence in the writing – alongside traditional research paper structure; 2. Analyses – a lot of great work here clearly, but again – its not communicated well – its not clear why you chose the model/s you did for example – as a reader I shouldn't need copious amounts of domain knowledge or to do to another paper to get background info – which I felt as though I did here. Similarly - how the analyses relate to specific pre determined hypotheses, this is missing – and alongside some slightly odd reporting of sensitivity analyses, raises question marks. 3. Importance and implementation – why is this research vital to field, how does it relate to psychosis – what does it mean for someone with psychosis? How does your highly specific question link back up to the wider world?

Thank you for your review of our manuscript and your suggestions. We provide point-by-point responses to each of your comments below.

General thoughts by section

Intro

- The link between psychosis symptoms and belief updating under uncertainty could be improved – the paper discusses previous computational modelling findings in relation to tonic/phasic differences, but it is not immediately clear to a reader e.g., how these differences may explain symptoms in psychosis (like delusions) – what does it mean to say that there is an increase of tonic and/or phasic arousal levels? (unnecessary belief updating/ perceive the world as volatile)

We have rewritten the entire Introduction section, now including a new paragraph hypothesising a link between altered arousal and the reasoning biases found in psychotic individuals (p.2):

“We reasoned that this adaptive recruitment of phasic arousal during dynamic belief updating may be disturbed in psychosis, and that this disturbance might link the tonic ‘hyper-arousal’ found in psychotic individuals to their characteristic reasoning biases. Specifically, because of a well-documented opponent interplay between tonic and phasic activity modes of central arousal systems, we assumed that increased tonic arousal levels should reduce phasic, task-evoked arousal responses during cognition. If phasic arousal responses fail to track the probability of hidden state changes (‘change-point probability’, CPP) or the agent’s uncertainty during belief updating as a result, this should reduce the dynamic evidence re-weighting in changing environments, altering the dynamic belief updating process overall and potentially leading to more perseverative, inflexible beliefs.”

We hope that this addition addresses your comment satisfactorily.

- Mention of ‘adaptive’ belief updating – but not clearly defined how the models/tasks that you use will get at adaptive aspects of belief updating – other references that could be useful are those from Mike Browning’s lab at Oxford., – integrates adaptation to volatility and pupillometry (although is in anxiety)

We used the term ‘adaptive’ to capture the fact that the optimal belief updating (i.e., combination of prior with new evidence sample) in our task depends on the local context – with increased weighting of evidence in moments of surprise (high change-point probability) and uncertainty (prior belief close to zero). Here, ‘non-adaptive’ belief updating computations are ones that always place equal weight on new evidence, regardless of the prior and its consistency with the new evidence – as is the case, for example, with perfect evidence accumulation.

We recognise that the above use of ‘adaptive’ may cause confusion because, as you allude to, the term is also often used to refer to behavioural adjustments to changes in environmental statistics, such as hazard rate (which do not occur in our experiment). For this reason, we have now changed the phrasing throughout our paper to primarily refer to ‘dynamic’ rather than ‘adaptive’ belief updating. We also now include a statement in the Introduction that specifically elaborates on what we mean by dynamic belief updating (p.2): “This normative process entails a dynamic ‘upweighting’ of new evidence (or, analogously, ‘down-weighting’ of prior beliefs) at moments when either the probability of a change in the hidden state, or uncertainty about the hidden state, are high.”

Thank you for flagging work by Browning and colleagues, which we agree is very relevant to our study. We have now cited a most immediately related paper (Browning, Behrens, Jocham, O’Reilly, & Bishop, 2015) in the Introduction.

- Some further elaboration of tasks and how they capture aspects of information processing could be useful

We have now included a new table (Supplementary Table 1) in the manuscript that describes all key task parameters and our motivation behind the specific choice of parameter values. We reference this table early on in the Results section (p.11), where we first provide a detailed description of the task, and again in the Methods (p.5). Please note that for succinctness, we have opted not to elaborate on details about the task in the Introduction.

- Uncertainty in this context is described only in relation to volatility. The first paragraph indicates “multiple sources of uncertainty” but this is not expanded upon or stated that the paper will only focus on volatility – links to the description of the normative account of learning (e.g., Piray & Daw, 2021a; pulcu and browning 2021)

Our task consists of two sources of uncertainty: that related to volatility in the generative state, and that related to noise in the generative distributions. Both of these sources of uncertainty determine the dynamics of normative belief updating on our task, as described in our earlier work (Murphy et al., 2021) and in our new Supplementary Table 1. You are correct that, in the original version of the Introduction, we only alluded to volatility as a source of uncertainty. We have corrected this in the new version, now stating (p.2):

“[Dynamic belief updating] is particularly challenging in natural environments, which mix multiple sources of uncertainty such as noise degrading the evidence obtained during a stable environmental state, and the possibility of hidden changes in the environmental state itself”.

- The intro talks about jumping to conclusion tasks, but while the task used here is similar, the analysis isn’t looking at accumulation of evidence (using drift diffusion methods e.g.,) but the individuals assessment of volatility. This might be useful info in the intro

Accumulation of evidence is, in fact, an essential part of the normative model for solving our task as well as the central focus of our behavioural analysis approach, as we elaborate in the following.

Behavioural modelling: The model we used (developed by Glaze and colleagues from Josh Gold's lab; Glaze, Kable, & Gold, 2015) is a generalization of the drift diffusion model for two-alternative forced choice decisions in non-stationary environments. In the extreme case of hazard rate $H=0$ (perfectly stable environment) the model reduces to the perfect (i.e., lossless) evidence accumulation postulated by the drift diffusion model; when $H=0.5$ (random environment), the model bases its choices only on the most recently encountered evidence sample (i.e., no evidence accumulation). In other words, if the subjective hazard rate parameter is (close to) zero in a given individual, we would actually fit the drift diffusion model for our task (for an interrogation protocol, rather than free response protocol; Bogacz, Brown, Moehlis, Holmes, & Cohen, 2006). Between these two extremes – where the actual hazard rate of our task resides – the model employs the dynamic, non-linear form of evidence accumulation that we describe in the manuscript. We agree that the individual assessment of the volatility is an important feature of the task – but this is also the central determinant of how evidence is accumulated (i.e., how it deviates from perfect evidence accumulation). Thus, we view these things as two sides of the same coin.

Model-free quantification of evidence accumulation: The assessment of “psychophysical kernels” (called “evidence weighting profiles” in the text using less technical language) that lie at the heart of our behavioral analyses are an established approach of directly quantifying the dynamics of evidence accumulation in a model-free fashion that does not make any assumptions about the decision computation (Waskom, Okazawa, & Kiani, 2019). The observation of significant weights on final choice of evidence samples at all positions in the sequence (Figure 2c) is a direct demonstration of protracted evidence accumulation. This general approach to behavioral data is in keeping with a growing impetus in computational cognitive and mathematical neuroscience to leverage precise psychophysical analysis tools to infer and quantify cognitive computations rather than blindly relying on fitted model parameters (Waskom et al., 2019). In combination with modelling, this approach can also identify diagnostic signatures of latent computations and falsify alternative models (Palminteri, Wyart, & Koechlin, 2017): The observation that the weight of evidence on choice is modulated by CPP and uncertainty (Figure 2d,e) provides critical support for the non-linear evidence accumulation process postulated by the normative model and qualitatively rules out linear alternative models such as drift diffusion or leaky accumulation. Such important nuances could not easily be inferred from the values of parameter estimates.

Your comment made us realise that this aspect was not sufficiently clearly spelled out in the previous version of the manuscript. We now make this point explicitly in the Results section (p.12).

- In general I feel the intro failed to explain and motivate the research - and importantly the gap in existing literature that this research is filling

As noted above, we have now rewritten the Introduction. We hope that the new version serves to better highlight the knowledge gap that our study is seeking to fill. We note in particular the following statements that we hope will be beneficial in this regard (p.2):

“It has been unknown if and how changes in arousal levels in individuals with psychotic traits relate to aberrations of probabilistic reasoning – in particular, to dynamic belief updating.

Here, we developed an integrated approach to close this gap, building on recent computational and cognitive neuroscience studies implicating central arousal systems in dynamic belief updating”.

Methods

- It was unclear to me how the “oversampling” approach was done. From the information given it could not be directly replicated – e.g., how many additional people did you sample to get to the amount needed, or are the authors talking about bootstrapping? – how many people ended up in each group? It says “aiming for 50% with a P-score above 50th percentile – does that mean that in the subsamples there were $n=45$ in low and $n=45$ in high?”

We apologize for the lack of clarity in our initial submission around our sampling approach, which we will explain in detail below. We have now clarified the procedure in the Methods section (p.3) and also acknowledge possible implications of our sampling approach in the Discussion (p.22).

To avoid overly strong skewness of P-scores with the majority scoring at the low end of the psychosis continuum, we aimed to recruit a sample that mirrored distributions found in large-scale community samples, where the 50th percentile equals a P-score of 1.4 (Schlier et al., 2015). An interim check of our sample data after recruiting 67 participants (excluding the 6 participants with incomplete data) revealed the low end of the continuum to be over-represented, with $n=40$ reaching a P-score that was less than or equal to 1.4 and only $n=27$ reaching a score that was higher than 1.4. To achieve the distribution that had been found in large-scale studies, we therefore adjusted our recruiting strategy. To this aim, we prescreened interested individuals for P-scores before inviting them to the lab. Out of 75 prescreened individuals, 36 scored higher than the P-score cutoff of 1.4 and were invited to participate, out of which 18 completed the study. Additionally, we invited 5 randomly chosen individuals from those with prescreened P-scores lower than 1.4 to participate, in order to fulfil the pre-calculated subsample size.

- The differences between “symptom” groups were still very small – feature of general population sample – a larger sample would be better (not a helpful comment I realize but perhaps can be reflected in the conclusions that are drawn from the research/future directions)

We have now included a passage in the Discussion (p.22) highlighting the fact that our observed effects were generally small, and that a key direction for future work will be to assess the degree to which our findings generalize to individuals with more pronounced symptoms (e.g. diagnosed ultra-high-risk status or full diagnosis of schizophrenia).

- Possibly the inclusion of the individual with psychosis in figure 3d results is misleading – I would suggest reporting in the results section too

We now report in the Results section (p.16) both the presence of this participant in the high P subgroup, and the effect of their removal on the pattern of results.

- Computational Modelling – a normative model was compared with a perfect match (ideal observer) model, and final evidence models i.e., recency. Model comparison (relative goodness of fit) was computed using BIC – where BIC including the n free parameters and trials

We differentiate in the paper between i) idealized decision processes (ideal observer, perfect accumulation, last-sample only) that we use only to compute *benchmark* accuracies against which to initially compare the responses of our human participants; and ii) fits of the normative model, where we fit free parameters of different versions of the normative model to the participants’ choices. What you refer to as ‘perfect match (ideal observer)’ and ‘final evidence’ models correspond only to the idealized decision processes, and we do not fit these to participants’ behaviour. Each of these idealized processes are in fact special cases of the

normative model that we *do* proceed to fit (specifically, ideal observer corresponds to a fit of the normative model with $\hat{H}=H=0.1$ and noise=0, while the last-sample only process corresponds to a fit of the normative model with $\hat{H}=0.5$ and noise=0). We have now clarified this distinction through additional introductory text in the part of the Results section where these are initially introduced (p.12).

o It says that 4 different model variants were fit -but I am unclear what these were exactly – are the results of the model comparisons reported? It directs the reader to supplementary fig 2 – but this doesn't clear things up for me

o The winning normative model includes 2 free parameters – the process of getting to this model configuration could be more clearly explained

We have now clarified our model fitting procedure: both by being more explicit in linking to the Methods section and Supplementary Figure 2a in the Results (p.12); and by unpacking in the Methods section (p.8) how we used a combination parameter recovery analyses and quantitative model comparison to determine the winning model.

The free parameters of the specific model variants that we fit are listed in Supplementary Figure 2a, and the meanings of each of the four possible free parameters are described in the Methods section (which we direct readers to when the model fitting is first introduced in the Results) entitled 'Main task model fitting and comparison'. We prefer to reserve the details of the model fitting for the Methods and the associated supplementary figure because we do not view them as essential to communication of the main results, and our preference is to not elongate an already long Results section.

- Recency/primacy effect – “psychophysical kernels” using logistic regression – and then computing the “kernel half difference”.

o The 2 sample t test used to compare groups not fully described -any covariates? Not enough information about the analysis here.

We have used non-parametric permutation tests throughout, so there are no 2-sample t-tests. For any analyses of the kernel half difference measure we do not include co-variables.

We have now rewritten the initial description of the kernel half difference measure, provided in the Results section on p.12: “We used a simple scalar metric to quantify the recency effect, by subtracting mean weights across the first half of the sequence (first 5 samples) from mean weights across the second half of the sequence (last 5 samples). Positive values of this so-called ‘kernel half difference’ metric indicated recency in evidence weighting, negative values indicated primacy, and values close to zero indicated equal weight to all samples.” To further clarify how the metric was computed, we also now mark the first and second sequence halves in Figure 3a and denote their subtraction to form the kernel half difference measure plotted in Figure 3b.

We hope that these changes will help resolve any possible confusion about our approach.

Results

Psychosis group agnostic results

- Behavioral analyses showed that all participants performed within expected limits but were close to ideal observer than other strategies. However – these results should be more clearly presented

We have now rewritten parts of the text corresponding to presentation of these results (p.12), and also now report specific proportions of the sample who performed better than the two sub-optimal decision processes that we considered (perfect integration, and last-sample only).

- At an individual level its suggested there was significant variation – but I am unclear about how this is tested or what the word significant refers to

Where we describe the individual differences in estimates of subjective hazard rates (\hat{H} ; p.12), we now replace the term “significant variation” with “substantial variation” in order to avoid the apparent confusion that this may refer to a test of statistical significance. No such test was conducted and this is intended to be a qualitative statement about the reported wide range of \hat{H} values around the true task H .

- The figure 2 legend could be improved upon – esp. from d onwards.

We have now elaborated on the description of panels c-e in the legend for Figure 2, specifically providing more detail on how the kernels are computed which we hope will aid with their interpretation.

- The authors calculate the change point probability and the “uncertainty” using the normative model – and compute a modulation analysis that uses regression weights to determine the temporal integration of evidence – i.e., recency and primacy effects – importantly comparing the first and last 5 trials. It would be useful to more clearly explain how this is modulation and/or adaptation when its only the first and last 5 trials being compared

There may be a misunderstanding about our analysis of the psychophysical kernels and their modulations by CPP and uncertainty, which we will try to resolve below. We have now clarified our approach by including additional explanatory text where this analysis is described in the Results section (p.13-14, 16), and further highlighting in Figure 3c the time windows (i.e., sample positions) used for averaging the CPP and uncertainty modulation weights.

First, you are correct that we assess the relative degree of primacy vs recency by comparing the kernel half-difference measure (i.e., difference in kernel magnitude between the first vs the last 5 sample positions; just to clear up a potential misunderstanding, *sample positions* are not the same as *trials*, since each trial contains a sequence of 10 samples). See above for interpretation of the kernel half-difference metric.

We use a different approach for summarizing the *modulations* of evidence weighting by CPP and uncertainty. These modulation profiles are (also in the model) often non-monotonic across time. Thus, here, we do *not* focus on (or contrast) the first 5 versus the last 5 samples: After establishing these modulatory effects at the group level, we proceeded to average regression weights across all sample positions that were significant in the group analysis. Sub-groups of participants with different P-scores were then compared on the basis of the resulting effects.

- The summary of findings in section “participants implement adaptive belief updating with non linear evidence weighting” is unclear

Thank you for pointing us to this issue. We have now removed this brief summary sentence, instead opting to rely on the summaries of individual results as they are reported in the preceding paragraphs and concluding this section with the statement that these results constitute a replication of our previous findings from the same task. We also now include a short introductory paragraph to this section in an effort to better frame it with respect to our previous work (p.11).

Results pertaining to psychosis symptoms

- “Grouped kernels into bins (quintiles) defined by the associated p-scores” - I think authors are saying they took a subset of the sample (highest and lowest) but it doesn’t say how many people this is – and how it relates to kernels...

A quintile refers to 20% of the entire distribution, which amounts to 18 participants per bin in our dataset. We now state this in the part of the Results section where we introduce the associated analyses (p.14). We have also rephrased the corresponding sentence somewhat to clarify the procedure: “To evaluate the relationship between psychosis proneness and the detailed shapes of the weighting profiles, we grouped participants into bins (quintiles; $N = 18$ per bin) defined by the associated P-scores and focused our comparisons on the two participant subgroups with the lowest and highest P-scores”.

- First authors report that change point trials – proportion correct – differs significantly between groups – supp material fig 4a shows this but doesn’t report effect sizes

We use non-parametric statistical tests throughout in order to avoid potentially problematic assumptions about the underlying data distributions. The test that you are referring to here is a two-sample permutation test. Well-established measures of effect size do not exist for this form of statistical test and we do not report such measures for this reason. However we do now accompany all permutation tests with 95% confidence intervals on the key statistics to give the reader additional information about the nature of the effects that we report.

o Also they are reporting both the comparison between groups (quintiles) and the correlation. Computing both analyses needs to be both better justified, and also should be clearly related to pre defined hypotheses clearly stated up top.

Our dual approach to analysing individual differences in our data was motivated by the fact that both categorical (e.g. diagnosis of schizophrenia or ultra-high risk status) and continuous (psychosis continuum) views on psychosis are common in the literature. While we did not have strong *a priori* hypotheses about which of the two lines of inquiry, if either, would yield clearer effects, we think that there is value in exploring both here. We now make this point early on in the Results section (p.11).

- the noise parameter differed between groups – supplementary fig 4d and h – decision noise increased in high group – relates to previous findings for decision noise (temperature parameters) in psychosis refs – experiment - Katthagen et al., 2018, review - Gibbs-Dean et al., 2023

Thank you for pointing us to these articles. We now cite the Gibbs-Dean et al. (2023) review, which we agree is highly relevant, when first reporting the observed relationship in our data between P-scores and decision noise. Katthagen et al. (2018; <https://journals.plos.org/ploscompbiol/article?id=10.1371/journal.pcbi.1006319>) does not seem to refer to the relationship between diagnostic status (schizophrenia vs healthy control) and an equivalent model parameter.

- subjective hazard rate – The way its written seems to suggest that the sensitivity analyses removing the individual with psychosis disorder is selectively applied to some analyses and not others. If this is not the case it needs to be better explained. If it is, this is problematic if not clearly justified beforehand (and perhaps even then)

We reran all associated main text analyses (i.e. all analyses contained in Figures 3-5) with the participant with a prior diagnosis of drug-induced psychosis removed. We now state this point in the Results section (p.16) as follows:

“We note here that the high P subgroup included one individual with a prior diagnosis of psychosis due to substance abuse (P-score = 2.1). However, with the exception of the combined CPP and $-|\psi|$ modulations (Figure 3d; $P = 0.1$), all effects in Figure 3, as well as Figures 4 and 5 below, remained statistically significant when this individual was excluded.”

Please note that the additional analyses reported in red text in Supplementary Figure 4c,g, of the relationship between P-scores and subjective hazard rates, pertain to a different participant entirely: a participant that is a possible statistical outlier in the distribution of subjective hazard rates. This participant was not identified as an outlier on any other measures of interest, therefore we only report results with them excluded from analyses of subjective hazard rate.

- Findings for the kernels and psychosis:

o So here authors talk about adaptive component to belief updating –Its described as the summed regression coefficients across change point, and uncertainty modulations. I was lost in this discussion, its is not that complicated I think but investment needed in clearer explanations

The rationale is that the sum of both modulatory terms is a single measure that captures the extent to which each individual implements the dynamic, non-linear evidence accumulation prescribed by normative model. We now state this explicitly on p.16 of the Results, as well as providing more detail on how this variable was computed in the same paragraph.

Discussion

- Unusual to start the first paragraph of the discussion solely with what previous findings have shown. I would be better to clear state your findings in the first paragraph of the discussion,

o – speaks to a wider problem about clearly stating the aims, hypotheses, and crucially the importance and implementation possibilities for this research –If the hypotheses were more clearly defined then it would be easier to relate the findings back

We have now restructured and rewritten large parts of the Discussion, including starting with a clear summary of our findings (p.18-19). In this same paragraph we also now link our findings specifically back to our hypothesis that the aberrant dynamic belief updating in psychosis proneness is associated with altered engagement of arousal systems.

- Lack of discussion about (lack of) findings for negative and depression. Similarly, where some findings are in line with wider literature e.g., psychosis and decision noise

We have now added a new paragraph to the Discussion highlighting the specificity of our findings to dynamic belief updating signatures (but not working memory performance) and the positive symptom dimension (but not negative or depressive dimensions; p.21). We also now state in the Discussion that our finding of a relationship between P-scores and decision noise is consistent with previous work (p.19).

- The findings aren't very clearly contextualized in relation to previous literature – after stating the findings clearly – how do this relate to other research e.g., linked back to psychosis literature and symptoms in details

As mentioned above, we have rewritten the Discussion and now attempt to better contextualize our findings in relation to existing work. This includes a new section on the relevance of our findings to predictive coding accounts of psychosis (p.21-22).

- The findings that are discussed are overstated for what (I think) the statistical results actually show.

As part of our rewrite of the Discussion, as well as the Abstract, we have toned down statements around the significance and implications of our findings.

-Does the paper represent an advance in understanding which may influence thinking in the field?

The authors suggest that they have replicated findings that have been shown in psychosis groups, in a “more general” population. Its not clear what the importance of this is – I think authors could improve on clearly stating the importance of the research, and how it further advance the field

In our rewritten Discussion we have attempted to communicate the importance of our findings and general approach more clearly, which constitutes more than simply replicating previous findings with psychosis groups in a community sample. Key points are (i) that the current work builds on our recent computational and physiological insights from the same task, providing a strong platform from which to connect to neural mechanism and implementation; (ii) our interrogation of specific, theoretically-motivated behavioural signatures of dynamic belief updating, which yielded insights that may have been obscured by a focus on static model parameter estimates alone; (iii) ways in which our approach overcomes limitations of the classic beads task that has generated many of said findings with psychosis groups; and (iv) our integration of behavioural findings with pupillometry, which further assists us in connecting with candidate neural mechanisms. Each of these points are emphasised in our Discussion.

-Does the article presents an original study, new analysis, new model, or a direct or extended replication of previous work?

Yes - While the research is partially aiming to replicate pervious work, there is originality in the methods – e.g., pupillometry + computational modelling in psychosis.

-Are the data and analysis technically sound? Are they appropriate to answer the research question, e.g., are causal research questions addressed on the basis of causal, rather than correlational evidence?

I found it hard to tell, as I felt the methods and analysis were not very clearly explained. The analysis was not clearly hypothesis driven, which affected my ability to assess it technically

We hope that the clarifications around methods, analysis and our hypotheses that we have attempted to provide following your above comments and suggestions serve to address these issues.

Does the paper provide strong evidence for its conclusions?

I think here improvements could be made. The sample is small, effect sizes appear small. The conclusions drawn are quite strong for what the results actually are, perhaps overstated somewhat.

As mentioned above we have endeavoured to tone down the strength of our conclusions, both in the rewritten Discussion section and the Abstract. We also now include a limitations section in the Discussion (p.22) that makes references to both our limited sample size and relatively weak effects and offers suggestions for future work that could address these limitations.

-Is the study question important to scientists for a sub-field of psychology?

I believe it is, but I think the authors need to think about how to better communicate this to the audience

We hope that the revisions that we have undertaken in response to your comments and suggestions serve to better communicate the importance of our study.

Are there any special ethical concerns arising from the use of animals or human subjects? No

- Was the study preregistered and if so, did the authors follow the preregistration? Not preregistered as far as I can tell

Reviewer #3 (Expertise: Computational Psychiatry and Psychosis):

Summary:

This study investigated the relationship between evidence accumulation and pupil dilation during a volatile belief updating task (i.e., with change points). Their rationale for the study was that previously identified increased tonic arousal in psychotic populations would likely drive reduced phasic arousal, which in turn may impact pupil responses. Because of the altered pupil responses patients may have altered evidence accumulation. The purpose of this study is to assess the link between phasic arousal, as measured by pupil responses, and evidence accumulation processes, as measured by behavioral and computational metrics. The study was conducted in healthy controls in line with the psychosis continuum framework.

The authors replicated previous results in this larger sample, finding that participants update their beliefs in this task consistent with the normative model presented in the paper. The model clearly fits the data well and best fit parameters can replicate participant behavior. The supplement contains model validation and comparison measures, consistent with best practices. There is also a good degree of interindividual variability across fitted parameters. The authors note in the methods that their recruitment strategy enriched the sample for higher P-score participants, which is often necessary for this kind of work to obtain the needed range of scores to test for interindividual differences.

To assess the effect of psychosis proneness, participants were grouped into quintiles by P-score and focused on group-comparisons of the highest and the lowest P-groups, but dimensional correlations in the supplement including all participants were consistent with these results. The primary findings were a positive relationship between P-score and the noise parameter across all participants (i.e. more noise associated with more psychosis proneness), general performance differences across the high and low P-group, and differences in belief updating dynamics.

In terms of pupil dynamics, the authors show weaker pupil-linked arousal for the high P group and that the degree of encoding is associated with both behavioral and computational measures of belief dynamics (Kernal Half Difference/ CPP+Phi kernal) across all participants.

This is a very nice paper. The task is well designed, the computational modelling is top-notch, and the pupillometry provides a novel contribution to the literature. I have a few comments that I hope will help the authors improve the manuscript. Great job!!

Thank you very much for your positive assessment of our work and your insightful suggestions, which we have now addressed.

Introduction

Page 2: "The normative process entails a dynamic 'upweighting' of new evidence at moments when the probability of a state change ('change-point probability', CPP) or uncertainty in the belief are high." It has also been suggested that there may be down-weighting of prior knowledge, which may produce similar results. For example: <https://www.ncbi.nlm.nih.gov/pmc/articles/PMC8041039/>

The authors may want to briefly address this distinction and why their study/intro focuses on upweighting of new information.

Thank you for raising this distinction. We have rephrased the statement in the Introduction that you highlight to now avoid preferentially emphasising upweighting of new evidence over downweighting of prior beliefs as candidate mechanisms for dynamic belief updating on our

task (p.2). Indeed, we view these as two sides of the same coin and not easily distinguishable in our data/model. We also now include a new paragraph in the Discussion (p.21-22) relating our findings to predictive coding accounts of psychosis, wherein we explicitly acknowledge that aspects of our findings are consistent with proposals that positive symptoms of psychosis may arise from overly strong/precise prior beliefs.

Page 2: “Thus, increased tonic arousal levels observed in psychosis may reduce the phasic, pupil-linked arousal responses tracking CPP or uncertainty during belief updating. This, in turn, may reduce the dynamic evidence reweighting in changing environments, altering the belief updating process overall.” I think it may be useful to clarify this sentence a little more and connect for the reader more why differences in pupil-linked arousal may reduce dynamic reweighting. I can see how this might work in a visual context, but since it doesn’t extend to other sensory domains, and considering that most hallucinations are auditory, I think some more context would help.

We have rewritten the Introduction and elaborated on our hypothesis relating arousal differences to altered belief updating in psychosis (p.2). We are now more explicit about the role of pupil-linked arousal in dynamic belief updating that has been identified in previous studies and how this informs our hypothesis. This includes that fact that the modulation of evidence weighting by pupil responses is supramodal (i.e. known to be present in both visual and auditory sensory modalities) – which we note here is expected given the global (diffuse projections not limited to one sensory modality) nature of the ascending arousal systems that we use pupil responses to infer the activity of.

Results

Page 6: “While the previously discovered evidence weighting effects were statistically significant at the group level, we also observed substantial individual differences in the evidence weighting profiles (Figure 2c-e).” Could you clarify the “Group” here, in “group-level”? Are you referring to the whole group, or the P-score groups?

We have rewritten this part of the Results in response to comments from other reviewers, and this sentence has been removed in the process. We initially meant to refer to the whole group as opposed to the distinct P-score groups.

Page 6 – 8: My only major question is related to the relationship between psychosis proneness and the computational parameters. The authors report a positive relationship between P-score and the noise parameter across all participants (i.e. more noise associated with more psychosis proneness) in Supplemental figure 4h. Given this, I was surprised to see that the high P group showed more primacy/less recency than the low P group. I would have expected that the noisier group would have more recency, since prior beliefs would be less reliable leading to more reliance on newer evidence. Some previous work has shown this and is at least consistent with the noise results you found (<https://pubmed.ncbi.nlm.nih.gov/36548395/>). In figure 2a the high P group (i.e. the noisy group) appears to update slower than the low P group, also consistent with the notion of unreliable priors (though it’s possible I am misunderstanding the y-axis). Could you speak to this seeming contradiction?

The noise parameter in our model specifically captures “decision noise”: that is, noise in the translation of the final posterior belief (after accumulating all samples on a trial) into categorical choice. A useful analogy is with the inverse temperature parameter commonly employed in, e.g., reinforcement learning models, in that the parameter here captures choice variability that is not otherwise explained by the sensory evidence and modelled beliefs (in RL, the modelled choice values); in other words, it captures the ‘randomness’ in choices when modelled

evidence/beliefs are accounted for. As such, the noise parameter is not clearly related to how the precision of prior beliefs might shape the weighting of new evidence as the trial unfolds.

Given the correspondence to unexplained choice variability, it is expected that the main effect of changes in our noise parameter will be to scale the overall *magnitude* of the psychophysical kernels. Intuitively: if noise is infinitely high and choices are not based on evidence/beliefs at all, the kernel should be zero for all sample positions; whereas if noise is toward zero and choices are strongly based on evidence/beliefs, the kernel weights should be of high magnitude. To illustrate this point, we simulated kernels from the ideal observer version of the normative model corrupted by different levels of decision noise and plot the results in Figure R1 below. This shows that noise has the effect of multiplicatively scaling the kernel weights while preserving kernel shape, consistent with the main effect on kernel magnitude. By contrast, the effect of changing the other free parameter in our model, the subjective hazard rate, will be to affect the kernel *slope*: lower hazard rates will produce flatter kernels, while higher hazard rates will produce kernels with steeper positive-going slopes (corresponding to greater recency in evidence weighting).

Figure R1. Simulated effect of changes in decision noise on psychophysical kernels. Kernels generated by the normative model using the true task hazard rate ($H=0.1$) and subject to seven different levels of decision noise. Lighter blue colours correspond to lower noise levels.

The simulations in Figure R1 show that changes in the noise parameter can affect our kernel half difference measure (with a larger difference expected for lower levels of noise). Thus, the kernel half difference effect we find could be driven by a *combination* of noise and subjective hazard rate effects. In a new control analysis, we replicate the kernel half difference effect after normalizing (per participant) the kernel half difference measure by the sum of all kernel weights, so that it is independent from kernel magnitude. This shows that the effect is not only explained by increased decision noise. We now report this additional analysis in the Results section (p.15).

Also, more generally, could you plot along with the data the best fit model predictions (i.e. a posterior predictive check) for figure 3. It's possible that I am just mis-modeling the task in my head and these model checks would clarify what is "supposed" to happen here.

We have now plotted the predictions of the best-fitting model alongside the data in Figure 3. Thank you for this suggestion and for your other comments, which have improved the manuscript.

References

- Bogacz, R., Brown, E., Moehlis, J., Holmes, P., & Cohen, J. D. (2006). The physics of optimal decision making: a formal analysis of models of performance in two-alternative forced-choice tasks. *Psychological Review*, *113*(4), 700-765.
- Browning, M., Behrens, T. E., Jochem, G., O'Reilly, J. X., & Bishop, S. J. (2015). Anxious individuals have difficulty learning the causal statistics of aversive environments. *Nature Neuroscience*, *18*(4), 590-596. doi:10.1038/nn.3961
- Gibbs-Dean, T., Katthagen, T., Tsenkova, I., Ali, R., Liang, X., Spencer, T., & Diederer, K. (2023). Belief updating in psychosis, depression and anxiety disorders: A systematic review across computational modelling approaches. *Neuroscience and Biobehavioral Reviews*, *147*, 105087. doi:10.1016/j.neubiorev.2023.105087
- Glaze, C. M., Kable, J. W., & Gold, J. I. (2015). Normative evidence accumulation in unpredictable environments. *Elife*, *4*. doi:10.7554/eLife.08825
- Katthagen, T., Mathys, C., Deserno, L., Walter, H., Kathmann, N., Heinz, A., & Schlagenhauf, F. (2018). Modeling subjective relevance in schizophrenia and its relation to aberrant salience. *PLoS Computational Biology*, *14*(8), e1006319. doi:10.1371/journal.pcbi.1006319
- Murphy, P. R., Wilming, N., Hernandez-Bocanegra, D. C., Prat-Ortega, G., & Donner, T. H. (2021). Adaptive circuit dynamics across human cortex during evidence accumulation in changing environments. *Nature Neuroscience*, *24*(7), 987-997. doi:10.1038/s41593-021-00839-z
- Palminteri, S., Wyart, V., & Koechlin, E. (2017). The importance of falsification in computational cognitive modeling. *Trends in Cognitive Sciences*, *21*(6), 425-433. doi:10.1016/j.tics.2017.03.011
- Schlier, B., Jaya, E. S., Moritz, S., & Lincoln, T. M. (2015). The Community Assessment of Psychic Experiences measures nine clusters of psychosis-like experiences: A validation of the German version of the CAPE. *Schizophrenia Research*, *169*(1-3), 274-279. doi:10.1016/j.schres.2015.10.034
- Waskom, M. L., Okazawa, G., & Kiani, R. (2019). Designing and interpreting psychophysical investigations of cognition. *Neuron*, *104*(1), 100-112. doi:10.1016/j.neuron.2019.09.016

23rd Aug 24

Dear Dr Murphy,

Your manuscript titled "Individual Differences in Belief Updating and Phasic Arousal Are Related to Psychosis Proneness" has now been seen by our reviewers, whose comments appear below. In light of their advice I am delighted to say that we are happy, in principle, to publish a suitably revised version in Communications Psychology.

We therefore invite you to revise your paper one last time to address the remaining concerns of our reviewers and a list of editorial requests. At the same time we ask that you edit your manuscript to comply with our format requirements and to maximise the accessibility and therefore the impact of your work.

EDITORIAL REQUESTS:

SUBMISSION INFORMATION:

OPEN ACCESS:

* **DATA AVAILABILITY:**

[link redacted]

Best regards,

Jennifer Bellingtier

Jennifer Bellingtier, PhD

Senior Editor

Communications Psychology

Eva R. Pool, PhD

Editorial Board Member

Communications Psychology

orcid.org/0000-0001-5929-1007

REVIEWER EXPERTISE:

Reviewer #1: Belief updating and Psychosis

Reviewer #2: Computational Psychiatry and Schizophrenia

Reviewer #2: Computational Psychiatry and Psychosis

REVIEWERS' COMMENTS:

Reviewer #1 (Remarks to the Author):

The authors have addressed my comments in this revision. I appreciate the edits very much.

Reviewer #2 (Remarks to the Author):

Thank you to the authors for all the work on this paper. All of the amendments suggested have been thoroughly dealt with or explained. On re-reading the paper with the amendments made I find it to be much improved. The alterations to the introduction, results, and discussion has made presentation of the findings more compelling and digestible. I would recommend this paper for publication.

Reviewer #3 (Remarks to the Author):

The authors have addressed all my comments and I am happy to recommend publication. Great job!